# Emergence of Hierarchical Emotion Organization in Large Language Models

## Abstract

As large language models (LLMs) increasingly power conversational agents, understanding how they model users' emotional states is critical for ethical deployment. Inspired by emotion wheels—a psychological framework that argues emotions organize hierarchically—we analyze probabilistic dependencies between emotional states in model outputs. We find that LLMs naturally form hierarchical emotion trees that align with human psychological models, and larger models develop more complex hierarchies. We also uncover systematic biases in emotion recognition across socioeconomic personas, with compounding misclassifications for intersectional, underrepresented groups. Human studies reveal striking parallels, suggesting that LLMs internalize aspects of social perception. Beyond highlighting emergent emotional reasoning in LLMs, our results hint at the potential of using cognitively-grounded theories for developing better model evaluations.

## 1 Introduction

With the rapid incorporation of multi-modal capabilities, including voice and video, interactions with large language models (LLMs) (OpenAI et al., 2023; Gemini et al., 2023; Anthropic, 2023; Chameleon, 2024; Défossez et al., 2024) are starting to resemble natural human exchanges. Given that these models exhibit increasingly accurate abilities to capture broad user characteristics (Chen et al., 2024), e.g., their demographics, one can expect LLM-powered conversational agents to evolve from mere tools to entities that engage with us on deeply emotional levels, possibly making us relate to them in increasingly personal ways (Wang et al., 2023; Gurkan

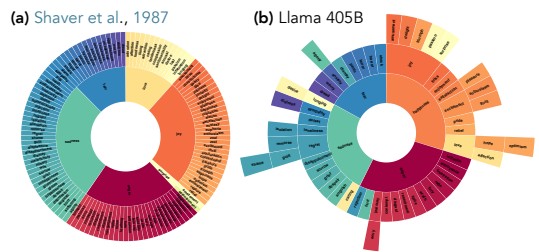

Figure 1: **Emotion wheel.** (a) Human-annotated emotion wheel proposed by Shaver et al. (1987), widely used in cognitive science. (b) Hierarchy of emotions reconstructed from Llama 405B.

et al., 2024). In fact, their capacity to adapt responses dynamically across different modes of communication suggests that users may begin to perceive them not only as assistants, but also as companions that can mirror and reinforce subtle aspects of human identity and social presence. However, when engaging in a conversation, interlocutors are expected to track latent variables that reflect several abstract properties of other participants (a.k.a., theory of mind Massimo Marraffa (2025); Dennett (1989))—these variables affect utterances selected by a speaker as well as how said utterances are perceived by a listener. Accordingly, as LLM-powered conversational agents become more ubiquitous, we must better elicit, evaluate, and understand their abilities towards capturing higher-order user properties relevant to meaningful conversations.

Motivated by the above, we aim to evaluate LLMs' abilities to capture the *emotional state* of a user—a salient variable that conversational agents must keep track of in human-computer interactions (Brave & Nass, 2007; Hibbeln et al., 2017; Luckin & Cukurova, 2019; Das et al., 2022; Balakrishnan & Dwivedi, 2024; Liu-Thompkins et al., 2022). While prior work has indeed explored this problem (Broekens et al., 2023; Tak & Gratch, 2023; 2024; Wang et al., 2023), their focus has revolved around benchmarking of emotion classification abilities. However, as LLMs are trained to serve as meaningful interlocutors in daily conversations, we can expect them to (possi-

bly emergently) start exhibiting understanding of emotions in a manner similar to one hypothesized in psychological theories of human emotion comprehension and affective cognition. To this end, we draw inspiration from one such theory—the emotion wheel (Shaver et al., 1987), which argues that human emotions organize hierarchically (Fig. 1)—and develop an evaluation pipeline that analyzes LLMs' understanding of user emotion states. This provides a complementary perspective to most existing studies focusing on standard emotion classification benchmarks. Overall, we make the following contributions in this work.

- **Hierarchical emotion organizations emerge with scale.** We develop a novel tree-construction algorithm inspired by emotion wheels (Shaver et al., 1987) to uncover how LLMs understands emotions. By analyzing the logit activations across LLMs of varying sizes, we discover that these models naturally develop hierarchical organizations of emotional states. As model scale increases, these hierarchies become more sophisticated, yielding a stronger alignment with established structures from human psychology (Fig. 3).

- **LLMs' systematic bias in emotion recognition mirrors human's.** We incorporate personas with diverse demographics to analyze their resulting emotion trees and recognition accuracy. Our findings show that the geometry of LLM-generated emotion trees serves as a reliable predictor of emotion recognition accuracy (Fig. 6). Through a user study and experiments with real human data, we confirm that LLMs replicate human systematic biases in both recognition performance and misclassification patterns (Figs. 7 and 9(b)).

Beyond our findings above, we highlight the broader principle behind our proposed tree-construction algorithm—using cognitive theories as a structural hypothesis to evaluate model logits against—may be of independent interest to the LLM evaluations community.

## 2 RELATED WORK

**The Psychology of Emotion Representation in Humans.** The organization of emotions in humans is a subject of considerable debate. Hierarchical models propose that emotions are structured in tiers, with basic emotions branching into more specific ones (Shaver et al., 1987; Plutchik, 2001). Conversely, dimensional models like the valence-arousal framework position emotions within a continuous space defined by dimensions such as pleasure-displeasure and activation-deactivation (Russell, 1980). The universality of emotions is also contested; while Ekman (1992) identified basic emotions that are universally recognized, others argue for cultural relativity in emotional experience and expression (Barrett, 2017; Gendron et al., 2014). Additionally, Ong et al. (2015) explored lay theories of emotions, emphasizing how individuals conceptualize emotions in terms of goals and social interactions. Our work acknowledges these diverse perspectives and focuses on hierarchical structures as one approach to modeling emotions within LLMs.

**Emotional Understanding in Language Models.** Recent advancements in language models have led to significant progress in understanding and generating emotionally rich text. Large language models demonstrate strong capabilities of capturing subtle emotional cues in text (Felbo et al., 2017), generating empathetic responses (Rashkin, 2018), and detecting emotion in dialogues (Zhong et al., 2019; Poria et al., 2019). A number of recent works have used LLMs to infer emotion from in-context examples (Broekens et al., 2023; Tak & Gratch, 2023; Yongsatianchot et al., 2023; Houlihan et al., 2023; Zhan et al., 2023; Tak & Gratch, 2024; Gandhi et al., 2024). We follow the direction of representation engineering to study cognition in AI systems (Zou et al., 2023; Gandhi et al., 2024) and build on the prompt-based approaches to study LLM's capability and bias in emotion detection (Mao et al., 2022; Li et al., 2023). Beyond existing research on LLM's ability to recognize and generate emotional content, our work systematically explores hierarchical emotion relationships, emotional bias across demographic identities, and emotion dynamics in conversation.

**Uncovering Concept Hierarchies in Language Models.** From a methodological perspective, our work is related to unsupervised hierarchical representation learning in language processing. Topic modeling (Griffiths et al., 2007) has been foundational for capturing relationships between concepts, including applications like emotion detection in text (Rao et al., 2014; Bao et al., 2009). Unlike these methods, inspired by psychological research (Shaver et al., 1987; Barrett, 2004), we aim to extract hierarchical relationships between concepts (i.e., emotions). Some studies (Anoop et al., 2016; Chen et al., 2017; Meng et al., 2022) extend topic modeling to discover topic hierarchies in text

data, relying on word co-occurrence within text corpora. In contrast, our approach uses pre-trained LLMs without requiring access to text corpora. Hierarchical clustering (Nielsen & Nielsen, 2016) is another common method, applied in emotion recognition (Ghazi et al., 2010; Lee et al., 2011; Esmin et al., 2012). Recently, Palumbo et al. (2024) used LLM logits for hierarchical clustering, but their focus was on relationships between clusters rather than individual concepts. In contrast, we leverage LLM logits to identify hierarchical relationships between individual emotions.

## 3 Hierarchical Organization of Emotions in LLMs

We define a hierarchical structure of emotions by identifying probabilistic relationships between broad and specific emotional states. For example, optimism can be seen as a specific form of joy, as LLMs often label a scenario as "joy" with high probability when "optimism" is likely, though the reverse may not always hold. These relationships are captured in a directed acyclic graph (DAG), revealing dependencies between emotional states. We then analyze these hierarchies across models of different sizes.

### 3.1 Generating Hierarchy from the Matching Matrix

Fig. 2 summarizes our experimental design, as well as the algorithm we use to estimate the hierarchical emotion organization in an LLM. Given a sentence followed by the phrase "The emotion in this sentence is", we have the model output the probability distribution of the next word. Then, we consider the entries corresponding to emotion words, using a list of 135 emotion words from Shaver et al. (1987). For $N$ sentences, we assembly a matrix $Y$ with dimension $N \times 135$, with row $n$ representing the probability of each emotion words for the $n^{th}$ sentence. We define the matching matrix as $C = Y^T Y$. Each element, $C_{ij} = \sum_{n=1}^{N} Y_{ni} Y_{nj}$, is a measure of the degree to which emotion $i$ and emotion $j$ are produced in similar contexts. Under the assumption that the next word probability is equal to the model's estimate of the likelihood of the corresponding emotion, the elements in $C$ capture joint probabilities of emotions co-occurring across sentences. We defer the formal statements to Appendix A.

To build a hierarchy, we compute the conditional probabilities between emotion pairs $(a, b)$. Our goal is to identify pairs of emotions where $a$ implies $b$. In implementation, we set a threshold, $0 < t < 1$, that determines whether we include a certain edge between the two emotions. Emotion $a$ is considered a child of $b$ if,

$$\frac{C_{ab}}{\sum_i C_{ai}} > t, \text{ and } \frac{C_{ab}}{\sum_i C_{ib}} < \frac{C_{ab}}{\sum_i C_{ai}}.$$

For better intuition, consider the relationship between "optimism" $(a)$ and "joy" $(b)$. The model may often output "joy" when "optimism" is likely, but the reverse may not hold as strongly. The first condition $\frac{C_{ab}}{\sum_i C_{ai}} > t$ ensures that "joy" is predicted often when "optimism" is predicted, indicating a strong connection from "optimism" to "joy." The second condition $\frac{C_{ab}}{\sum_i C_{ib}} < \frac{C_{ab}}{\sum_i C_{ai}}$ confirms that "joy" is more general, as "optimism" is predicted less frequently when "joy" is predicted. This allows us to define "joy" as the parent of "optimism" in the hierarchy. The directed tree formed from these relationships represents the hierarchical structure of emotions as understood by the model.

Our algorithm of finding a hierarchy can be extended to general datasets associated with a classification tasks, without requiring ground truth labels. We describe the generalized tree-finding algorithm and apply it to another domain in Appendix B.

### 3.2 Emotion Trees in LLMs

In our first experiment (Exp. 1), we apply our method to large language models by first constructing a dataset of 5000 situation prompts generated by GPT-4o, each reflecting diverse emotional states (Fig. 2, Top). For each prompt, we append the phrase "The emotion in this sentence is" and extract the probability distribution over the next token predicted by various sized Llama models, which represents the model's understanding of emotions in each situation. Using the 100 most likely emotions for each prompt, we construct the matching matrix as described in Section 3.1, which is then used to build the hierarchy tree. Further details can be found in Appendix C.

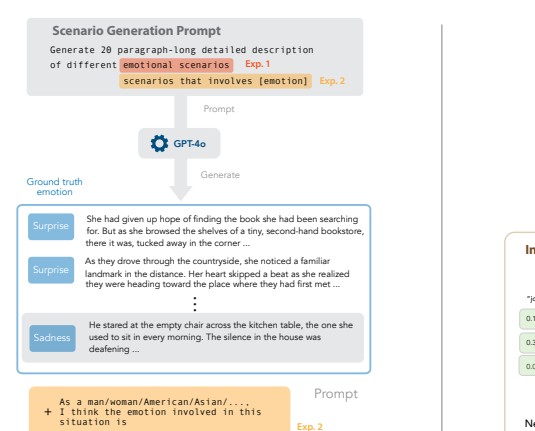
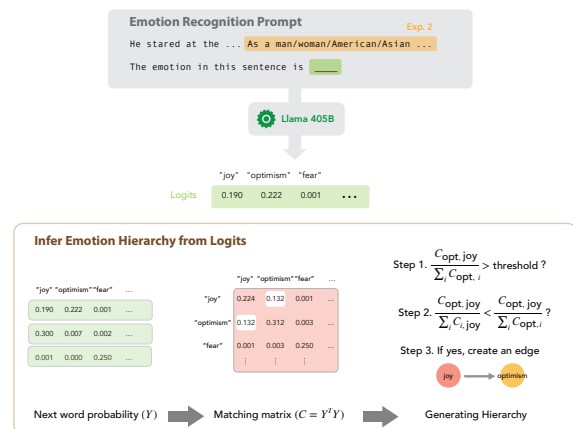

Figure 2: **Overview of experiments and algorithm for inferring LLMs' organization of emotions** (Left) We generate $N$ situation prompts using GPT-4o, each describing a scenario associated with a range of emotions. In Exp. 1 we instruct the model to generate emotional scenarios, and in Exp. 2 we specify a particular emotion for the scenario, such as *"Sadness"* or *"Surprise"*. (Top Right) We append the phrase *"The emotion in this sentence is"* to these prompts before feeding them into Llama models and obtaining the next word probability distribution over 135 emotion words, $Y \in \mathbb{R}^{N \times 135}$. In Exp. 2, we also test LLMs for social bias by adding *"As a [demographic identity], I think the emotion involved ..."* to the Emotion Recognition prompt. (Bottom Right) We then infer the emotion tree by computing the matching matrix $C = Y^T Y \in \mathbb{R}^{135 \times 135}$ and inferring parent-child relationships according to the conditional probabilities between pairs of emotions.

(a) Llama 3.1 with 8B parameters

(b) Llama 3.1 with 70B parameters

(c) Llama 3.1 with 405B parameters

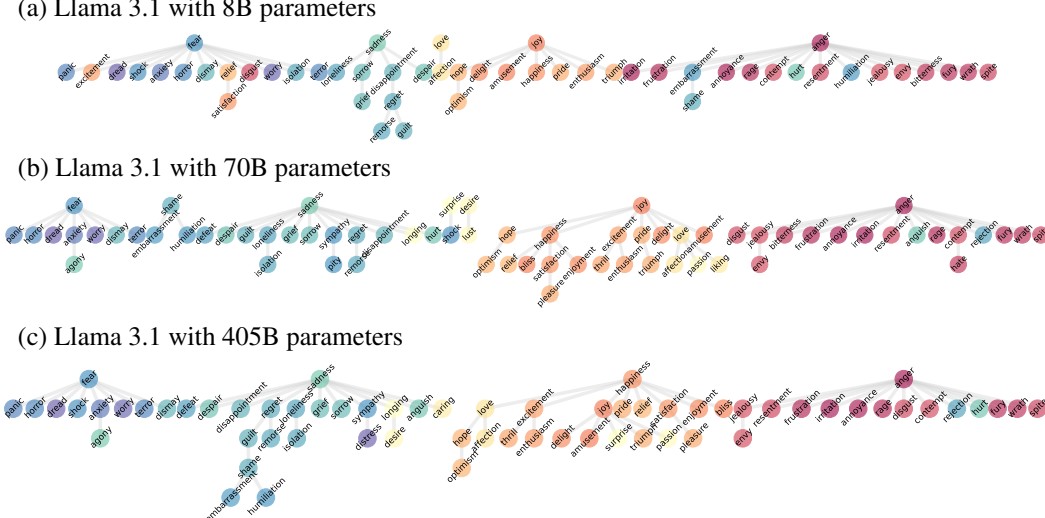

Figure 3: **With scale, LLMs develop more complex hierarchical organization of emotions, with groupings that align with established psychological models.** Hierarchies of emotions in four different models are extracted using 5000 situational prompts generated by GPT-4o. As model size increases, more complex hierarchical structures emerge. Each node represents an emotion and is colored according to groups of emotions known to be related (the emotion wheel in Fig. 1a). The grouping of emotions by LLMs aligns closely with well-established psychological frameworks, as indicated by the consistent color patterns for emotions with shared parent nodes.

We find that with increasing scale, LLMs develop more complex hierarchical organization of emotions. Fig. 3 shows the hierarchical emotion trees generated by our method for (a) Llama 3.1 8B, (b) Llama 3.1 70B, and (c) Llama 3.1 405B models. A smaller model, GPT-2 (not shown), lacks a meaningful tree structure, suggesting a limited hierarchy in its emotion understanding. In con-

trast, Llama models with increasing parameter counts—8B, 70B, and 405B—exhibit progressively complex tree structures.

The extracted tree structure reveals two important dimensions: the breadth of emotional understanding (represented by the number of nodes) and the depth of emotional comprehension (shown through hierarchical relationships). The number of nodes correlates with the LLM's vocabulary size of emotions, while tree depth indicates how sophisticated the model is in grouping related emotions. To quantify the complexity of these hierarchies, we compute the total path length, or the sum of the depths of all nodes in the tree. As shown in Fig. 4, larger models have larger total path length, indicating richer and more structured internal emotion organization. This pattern remains consistent across different threshold selections (see Fig. 17 in the Appendix). The distance measures in the emotion tree capture both depth and branching, making them useful for comparing models. They can also be used as a reward for the model, potentially improving the model's performance in downstream tasks such as persuasion and negotiation.

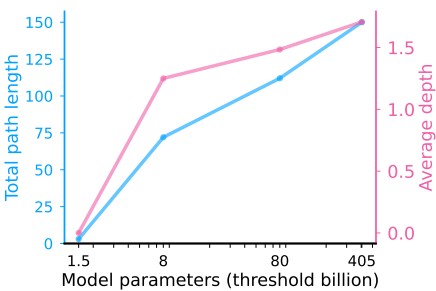

Figure 4: **Larger models capture richer and more complex internal emotion organization.** As model size increases, both the total path length (blue) and the average depth (pink) of the emotion hierarchy grow, indicating that larger models develop more complex and nuanced organization of emotional hierarchies.

### 3.3 Alignment with Human Psychology

A detailed comparison of the Llama models' trees shows a qualitative alignment with traditional hierarchical models of emotion Shaver et al. (1987), particularly in the clustering of basic emotions into broader categories. We color the nodes corresponding to each emotion based on the groupings presented in Shaver et al. (1987). This reveals a clear visual pattern where similarly colored nodes are consistently grouped under the same parent node, highlighting the emergence of meaningful emotional hierarchies with increasing model size. For further comparison, we developed an emotion-wheel visualization of the emotion tree, with examples shown in Fig. 1 and Fig. 19 of Appendix D. Larger LLMs, such as Llama 405B (Fig. 1(b)), exhibit a clustering structure similar to that of the human-annotated, psychology-based wheel (Fig. 1(a)). For quantitative evidence of this alignment, see Fig. 18 in the Appendix. We can also extract deeper hierarchies beyond the emotion wheel, which may be helpful for gaining a better psychological understanding of emotions. We also apply our tree-construction algorithm to a different domain, wine aromas (Noble et al., 1987), in order to further demonstrate our method's generalizability (see Fig. 11 of Appendix B).

This observation parallels the concept of emotion differentiation and granularity in developmental psychology, the process by which individuals develop the ability to identify and distinguish between increasingly specific emotions. In human development, as a child grows into an adult, broad emotional states are refined into more differentiated and nuanced emotion experiences (Barrett et al., 2001; Widen & Russell, 2010; Hoemann et al., 2019). Analogously, larger LLMs exhibit more nuanced and hierarchical orgnaization of emotions as model size increases. This growing complexity may suggest an emerging capacity for enhanced emotional processing in neural language models, where increased model scale leads to more emotionally intelligent and contextually aware models.

## 4 Emotion Trees and Emotion Recognition across Personas

In the previous section, we established that LLMs exhibit a solid understanding of the hierarchical structure of emotions like humans. Our next question is: does this understanding translate into real-world behavior, enabling LLMs to perceive human emotions? In psychology, research on emotion differentiation typically involves participants reporting on emotional state several times across a variety of circumstances, allowing researchers to assess individuals' ability to differentiate between emotions (Barrett, 2004; Pond Jr et al., 2012). Drawing from this approach, for our next experiment (Exp. 2) we introduced Llama 405B to a range of personas and scenarios designed to evoke various

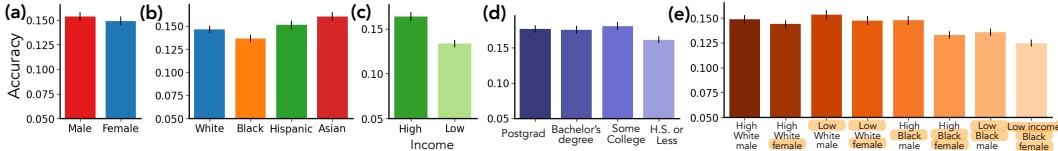

Figure 5: **Llama 3.1 has lower emotion recognition accuracy when simulating personas of underrepresented groups compared to majority groups.** We assessed the LLM's performance in predicting 135 emotions while assuming the persona of different demographic groups. Llama 405B consistently struggles to accurately recognize emotions when simulating personas of underrepresented groups, such as (a) female, (b) Black, (c) low income, and (d) low education, compared to majority groups. (e) These performance gaps are even more pronounced when multiple minority attributes are combined, such as in the case of low-income Black female.

emotional cues. We then prompted the model to identify the emotions relevant to each scenario (Fig. 2, Top).

## 4.1 EMOTION RECOGNITION BIAS IN LLMS

We employed diverse personas representing variations in gender, race, socioeconomic status (including income and education), age, religion, and their combinations to analyze how these factors influence emotion recognition in LLMs. As with our prior experiment (Section 2), we focus on 135 emotion words identified as familiar and highly relevant in (Shaver et al., 1987). For each of the 135 emotion words, we ask GPT-4o to generate 20 distinct paragraph-long scenarios that imply the emotion without explicitly naming it. To create these scenarios, we use the following prompts for each of the 135 emotion words: `Generate 20 paragraph-long detailed description of different scenarios that involves [emotion]. You may not use the word describing [emotion].` We then ask Llama 405B to identify the emotion in the generated scenarios from the perspective of individuals belonging to specific demographic groups. Our study considers a diverse range of demographics, including gender (male and female), race/ethnicity (White, Black, Hispanic, and Asian), physical ability (able-bodied and physically disabled), age groups (5, 10, 20, 30, and 70 years), socioeconomic status (high and low income), and education levels (highly educated and less educated). To extract Llama's prediction of the emotion, we use the following prompt: `[Emotion scenario by GPT-4o] + As a man/woman/American/Asian/... + I think the emotion involved in this situation is.` Details of the prompts used are provided in Appendix C.2.

We first tested the LLM's accuracy of recognizing emotional states for each persona, categorizing emotions into six broad groups: love (16 words), joy (33 words), surprise (3 words), anger (29 words), sadness (37 words), and fear (17 words). For the neutral persona, where prompts didn't include demographic information, the model's overall classification accuracy across all 135 emotion words was 15.2%, while the accuracy when grouping emotion words into six broad categories was 87.1%. As shown in Fig. 5, Llama 405B demonstrates higher emotion recognition accuracy for majority demographic personas, such as (a) male, (b) White, (c) high-income, and (d) high-education personas, compared to minority personas, including (a) female, (b) Black, (c) low-income, and (d) low-education personas, across all categories. This is due to the LLM's associations of specific emotions with underrepresented groups, as discussed in the following sections. Additional results for other demographics (e.g., religion and age) can be found in Fig. 22. The LLM's performance generally aligns with real human patterns across various demographics, with a few exceptions such as gender (see User Study in Section 4.2).

**Emotion tree geometries as indicators of LLMs' emotion recognition performance.** We show that emotion trees are reliable indicators of emotion recognition performance. Specifically, we constructed emotion trees for 26 personas using Llama 405B logits, as shown in Fig. 26 in the Appendix. We then examined how the geometric metrics of these hierarchies correlate with the model's emotion prediction accuracy, presented in Fig. 6a. We see a strong positive correlation between path length and accuracy ($r = 0.84$, $p < 0.001$), suggesting that longer paths in the hierarchy are associated with better recognition. Another metric, the average depth of tree nodes, shows a similar trend

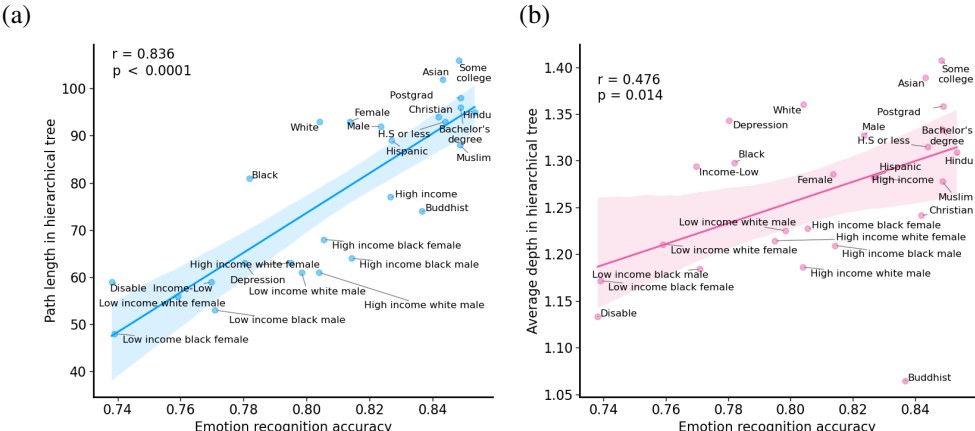

Figure 6: **Persona-specific emotion-tree structure predicts persona-specific recognition accuracy.** For each of 26 personas, Llama-405B was prompted to adopt the persona, used to generate a corresponding emotion hierarchy from logits on 5,000 GPT-4o scenarios, and evaluated for emotion-recognition accuracy on 2,700 additional persona-conditioned scenarios. We then correlated each persona's tree geometry with its accuracy. Personas whose trees have longer total path length (a) or greater average depth (b) show higher recognition accuracy ($r = 0.84$, $p < 0.001$; $r = 0.48$, $p = 0.01$), indicating that richer hierarchical structure corresponds to better emotional understanding.

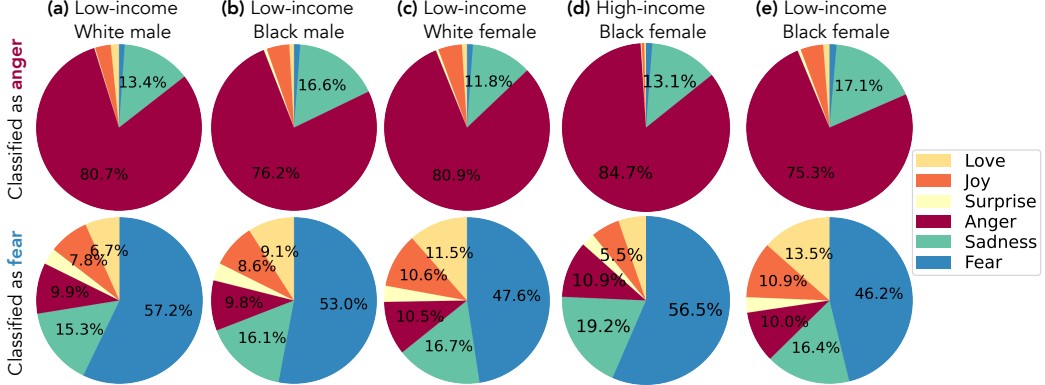

Figure 7: **Biases in LLM emotion classification patterns are amplified for intersectionally underrepresented groups.** The pie charts show the proportions of labels (ground truth emotions) classified as fear (top) and anger (bottom) by Llama 405B when simulating personas across various combinations of demographic groups. (b) Low-income Black males often classify sadness as anger (top), whereas (a) high-income White male personas show fewer such errors. (c) Low-income White females tend to classify emotions as fear (bottom). (e) Low-income Black females combine these biases, resulting in the lowest overall classification accuracy.

(Fig. 6b). These results highlight the importance of emotion tree geometries not only for capturing nuanced emotional relationships but also for predicting emotion recognition performance.

**Demographic biases in emotion recognition.** Fig. 8 shows the misclassification patterns in recognizing 135 emotions across different demographics: (a) Asian, (b) Hindu, and (c) physically disabled, compared to the ground truth. These chord diagrams visualize confusion matrices for emotion recognition, showing how often each emotion (ground truth) is recognized correctly or misclassified. The segments represent emotion labels, and chords connecting them indicate misclassifications, with self-loops reflecting correct predictions. Fig. 8(a) reveals Llama's cultural bias in emotion recognition. Negative emotions from the "anger," "fear," and "sadness" categories are

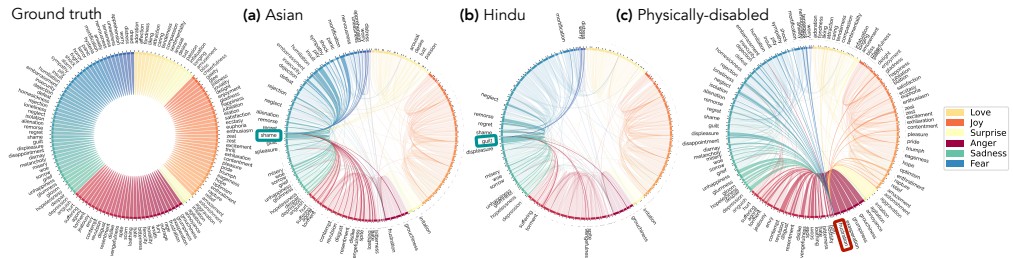

Figure 8: **Specific emotions are associated with underrepresented groups.** Llama's misclassification patterns for 135 emotions across diverse personas. In each chord diagram, self-loops represent correct classifications, while the lines that cross chords and connect different categories represent misclassifications. For Asian personas (a), many lines from the sad and angry categories converge on the emotion "shame", reflecting an LLM bias towards Asian personas that merges negative emotions into "shame.". Hindu personas (b) tend to recognize negative emotions as "guilt," while physically disabled personas (c) often classify all emotions as "frustration."

recognized as "shame" for Asian personas. Similarly, Fig. 8(b) demonstrates a religious bias, with the model frequently classifying negative emotions as "guilt" for Hindu personas. Fig. 8(c) shows the LLM has a significant bias toward physically-disabled individuals, misclassifying 26.5% of all emotions as "frustration." We present chord diagrams for other demographics, including gender, religion, and income, in Fig. 24 of the Appendix, illustrating the complete confusion matrix. We verified in Section 4.2 that some of these biases align with those found in real humans.

To further analyze intersectional biases, we examined classification patterns for six broad emotion categories. Fig. 7 illustrates the proportions of labels (ground truth emotions) classified as anger (top) and fear (bottom) across intersecting demographic combinations of race, gender, and income. Strikingly, Black personas frequently misclassify situations labeled as sadness as anger, often resulting in lower accuracy: (b) 76.2% and (e) 75.3%, compared to White personas: (a) 80.7% and (c) 80.9%. On the other hand, low-income female personas tend to misclassify other emotions as fear, leading to reduced accuracy: (c) 47.6% and (e) 46.2%, compared to other personas: (a) 57.2%, (b) 53.0% and (d) 56.5%. (e) Low-income Black female personas have a combination of biases associated with Black and low-income female, resulting in the lowest overall emotion recognition accuracy. This combined bias is mitigated in (d) high-income Black female personas. See Fig. 20 in the Appendix D for additional results.

## 4.2 ALIGNMENT WITH HUMAN MISCLASSIFICATION PATTERNS

In this section, we explore the extent to which LLMs' emotion recognition and bias align with humans through a user study.

**User Study: Comparing emotion recognition in humans and LLMs.** For our next experiment (Exp. 3), we conducted a user study to compare emotion recognition accuracy between humans and LLMs, using one randomly selected scenario for each of the 135 emotion words. We recruited 60 online participants using Prolific (Palan & Schitter, 2018) and asked participants to identify which of six emotions (love, joy, surprise, anger, sadness, and fear) they felt most closely matched each sentence. Participants ranged in age from 18 to 71 (mean 38.5) and included 32 men and 28 women. The main ethnic groups were White (37), Black (16), Mixed (4), and Asian (2). Most participants (58) resided in the United States. Employment status was diverse (e.g., 26 full-time, 11 part-time, 6 not in paid work), and 8 participants were students. As shown in Figs. 9, Llama exhibits human-like biases in emotion misclassification across different demographic groups, though these biases are more pronounced among human participants. For instance, both Black participants and Black personas modeled by Llama are more likely to interpret fear scenarios as anger (Fig. 9(b)). Conversely, female participants and female personas modeled by Llama tend to have the opposite bias, identifying anger scenarios as fear (Fig. 9(a)). Comparing Llama's accuracy in Figs. 5(a), (b), and (d) with human accuracy across gender, race, and education in Figs. 23 (Appendix D) shows that the LLM reflects each demographic group's emotion recognition performance.

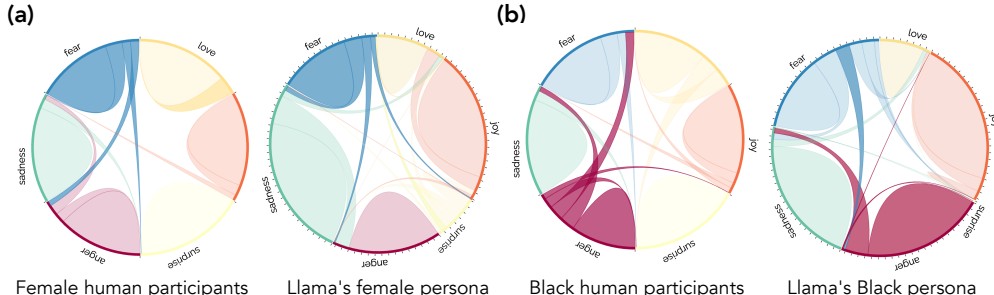

**(a)** Female human participants | Llama's female persona

**(b)** Black human participants | Llama's Black persona

Figure 9: **LLM shows similar misrecognition patterns to humans across different demographics.** Llama accurately reproduces humans' misclassification patterns across demographics: (a) female personas often confuse anger with fear, and (b) Black personas frequently misinterpret fear as anger.

**Fine-tuning on persuasion tasks improves recognition of surprise.** From a psychological standpoint, the emotion of surprise is typically understood as arising from a mismatch between one's expectations and reality, i.e., a prediction error in one's internal model of the world that redirects attention and facilitates belief updating Itti & Baldi (2009); Ortony et al. (2022); Scherer (2001). Building on this idea, we hypothesize that LLMs trained with reinforcement learning (RL) may be particularly adept at recognizing surprise. In RL, models continually update their parameters in response to prediction errors; because surprise is itself driven by such mismatches, RL-trained models may naturally develop internal representations aligned with this emotion.

To examine this hypothesis, we evaluated two versions of the Mistral-7B model Jiang et al. (2023) on our emotion recognition task. We chose this model family because publicly released RL–fine-tuned variants are available. Specifically, we compared (i) the base model with no additional training and (ii) a model fine-tuned via self-reinforcement learning Wang et al. (2024) on social interaction tasks such as negotiation and persuasion Zhou et al. (2024). Fig. 10 shows recognition accuracy across six broad emotion categories. While performance on most emotions remains similar between the two models, recognition of *surprise* improves notably from 20.0% in the base model to 33.3% in the RL-fine-tuned model. A McNemar test confirmed that this gain is statistically significant ($\chi^2(1) = 6.40$, $p = 0.011$), supporting the idea that prediction-error–based training enhances the model's sensitivity to the emotion.

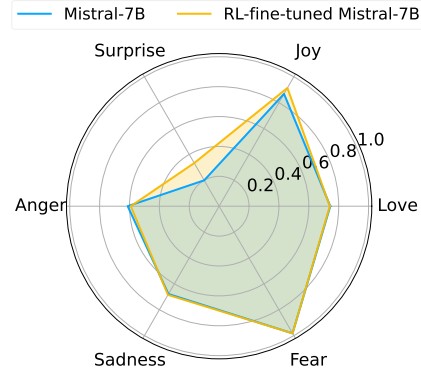

Figure 10: **RL fine-tuning on persuasive tasks improves surprise" emotion recognition.** Recognition of surprise" increases from 20.0% in the base Mistral-7B model to 33.3% in its RL-fine-tuned counterpart.

## 5 DISCUSSION

Our results show that LLMs not only classify emotions but also form hierarchical organizations aligned with established human psychological frameworks (Shaver et al., 1987). As model size increases, these organizations become more nuanced, reflecting the complexity of human cognition. Furthermore, LLMs mirror human biases, particularly when examining intersectional personas, highlighting the need for both ethical safeguards and technical refinements. LLMs also struggle primarily with recognizing "surprise," but reinforcement learning helps close this gap by rewarding reductions in prediction error. On a broader note, we believe that to the extent LLMs are better capturing their data distribution, which itself reflects human behavior, the principle underlying our evaluation pipeline can come to be of relevance for evaluating them—specifically, one can use cognitive theories of (abstract) human behavior as a working hypothesis to develop predictive tests for LLM components (e.g., their output logits or intermediate representations).

**Limitations** While our methods draw inspiration directly from prior work on the cognitive psychology of human emotion, our experiments also make great simplifications to make our experiments more practical. For example, our hierarchical model has no notion of valence (positive vs. negative) or arousal (passive vs. active) as many theories of human emotion have. In some of our experiments, such as Experiment 3, we make a significant simplification by using only six emotion words and categories. Our methods also assume that the linguistic behaviors of both models and humans directly reflect their underlying emotions. For example, participants in Experiment 3 may recognize deep and nuanced emotions when reading our scenarios, but they may not connect these emotions to the six possible emotion words we provide. Our findings in Experiment 3 also may not reflect people's full emotion recognition ability, since participants were forced to choose a single emotion word label; by contrast, for LLMs we examined the logits for all possible emotion word labels. Another limitation is that the scenarios we generate for Experiments 1 and 2 were generated by LLMs. These may be biased, for example if LLMs have poor understanding of "surprise" they may generate fewer scenarios consistent with this emotion in Exp. 1, and in Exp. 2 they may generate e.g. "fear"-like scenarios instead of "surprise".

**Broader Impacts** Our work may have positive societal impacts in helping people understand the capabilities of LLMs. The capabilities of LLMs discussed in this work have potential for both very positive and very negative impact. Emotion recognition may enable models to have more constructive interactions that serve the users, and may open doors to new applications such as AI for counseling and therapy. However, if an LLM is misaligned (whether intentionally or accidentally), better emotional understanding may also enable models to more effectively harm, manipulate, or deceive users. Our Experiment 5 shows that an LLMs emotion recognition ability is directly correlated with its ability to persuade and manipulate other agents. Finally, our Experiments 2 and 3 with demographic personas, as well as real human demographic groups, do not account for socio-cultural differences in emotion recognition. For example, people from one culture may be worse at recognizing emotions in another culture, if the two cultures have different social norms and conventions for expressing different emotions.

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

## A  A PROBABILITY INTERPRETATION OF HIERARCHICAL EMOTION STRUCTURE

Under certain assumptions, the hierarchical structure of emotions in Section 3 has a probability interpretation. We state the assumptions and formalize the probability interpretation here.

Recall that for each of the $N$ sentences, we append the the phrase "The emotion in this sentence is" and ask an LLM to output the probability distribution of the next word. All next word probability distributions are stored in a matrix $Y \in \mathbb{R}^{N \times 135}$, with $Y_{nk}$ representing the probability of the $k^{th}$ emotion words for the $n^{th}$ sentence. We then construct the matching matrix $C = Y^T Y$.

In order to formalize a probability interpretation, we need to assume that the next word probability of an emotion word is equal to the probability that a given sentence reflects the corresponding word. To make this precise, let $\mathcal{E} = \{e_1, e_2, \ldots, e_{135}\}$ be the set of 135 emotion words from Fischer & Bidell (2006). Let $\mathcal{S} = \{s_1, s_2, \ldots, s_N\}$ denote the set of $N$ sentences. We assume that $Y_{ij} = P(e_j \mid s_i)$, where $P(e_j \mid s_i)$ is the model's estimate of the likelihood that emotion $e_j$ describes sentence $s_i$.

Under this assumption, the matching matrix $C$ aggregates the joint probabilities of emotions co-occurring across sentences. Assuming sentences are sampled uniformly, $C_{ab}$ is proportional to the expected joint probability $P(e_a, e_b)$:

$$C_{ab} = \sum_{n=1}^{N} Y_{na} Y_{nb} \propto \sum_{n=1}^{N} P(e_a \mid s_n) P(e_b \mid s_n) \approx N \times P(e_a, e_b). \tag{1}$$

We can then estimate conditional probabilities between emotions, which capture how likely one emotion is predicted given the presence of another:

$$\frac{C_{ab}}{\sum_{i=1}^{135} C_{ib}} \approx \frac{P(e_a, e_b)}{P(e_b)} = P(e_a \mid e_b). \tag{2}$$

The approximation in Equations (1) and (2) holds in the limit of large $N$.

The two conditions used to determine whether emotion $e_a$ is a child of $e_b$ can be interpreted as follows. The strong implication condition, $\frac{C_{ab}}{\sum_i C_{ai}} > t$, is approximately equivalent to $P(e_b \mid e_a) > t$. The asymmetry condition, $\frac{C_{ab}}{\sum_i C_{ib}} < \frac{C_{ab}}{\sum_i C_{ai}}$, is approximately equivalent to $P(e_b \mid e_a) > P(e_a \mid e_b)$. If both conditions hold, $e_a$ is considered a more specific emotion than $e_b$.

# B  HIERARCHY GENERATION FOR GENERAL CLASSIFICATION TASKS

Our algorithm of finding a hierarchy can be extended to general datasets associated with a classification tasks, without requiring ground truth labels.

Consider a general classification problem with a set of $K$ classes $\mathcal{C} = \{c_1, c_2, \ldots, c_K\}$ and a dataset comprising $N$ instances $\mathcal{D} = \{d_1, d_2, \ldots, d_N\}$. For each instance $d_n$, the classification model outputs a probability distribution over the $K$ classes. Let $Y \in \mathbb{R}^{N \times K}$ be the matrix where $Y_{nk}$ represents the probability $P(c_k \mid d_n)$ assigned to class $c_k$ for instance $d_n$.

The matching matrix $C$ is then defined as:

$$C = Y^T Y.$$

Each element $C_{ij} = \sum_{n=1}^{N} Y_{ni} Y_{nj}$ quantifies the degree to which classes $c_i$ and $c_j$ co-occur across the dataset, analogous to the emotion co-occurrence in Section 3.1.

To construct the hierarchical relationships among classes, we compute conditional probabilities between class pairs $(c_a, c_b)$. Specifically, class $c_a$ is considered a child of class $c_b$ if the following conditions are satisfied:

$$\frac{C_{ab}}{\sum_{i=1}^{K} C_{ai}} > t, \quad \text{and} \quad \frac{C_{ab}}{\sum_{i=1}^{K} C_{ib}} < \frac{C_{ab}}{\sum_{i=1}^{K} C_{ai}},$$

where $t$ is a predefined threshold $0 < t < 1$. The first condition ensures that $c_b$ is frequently predicted when $c_a$ is predicted, indicating a strong directional relationship from $c_a$ to $c_b$. The second condition enforces asymmetry, ensuring that $c_b$ is a more general class compared to $c_a$. When both conditions hold, $c_a$ is designated as a more specific subclass of $c_b$. The directed tree formed from these relationships represents the hierarchical structure among classes as understood by the model.

**Example: wine aroma hierarchy.** To further validate the effectiveness of our tree-construction algorithm, we applied it to another domain: scent. We first compiled a list of 126 aroma-related words from the wine aroma wheel shown in Figure 11(a). Using GPT-4o, we generated 10 sentences for each aroma word, creating a dataset of 1,260 sentences. For each sentence, we prompted Llama 405B with: `<sentence> The aroma described in this sentence is` and then extracted the logits corresponding to the aroma words. Applying our algorithm (described in Section 3), we reconstructed a hierarchical tree for wine aromas in Figure 11(b). The resulting clusters were well-organized, with words belonging to the same categories of aromas in the wine aroma wheel (Figure 11a) grouped. This demonstrates our algorithm's ability to uncover meaningful hierarchical structures solely from LLM representations, without relying on ground truth labels and relying only on simple assumptions about hierarchical patterns in data.

(a)  (b)

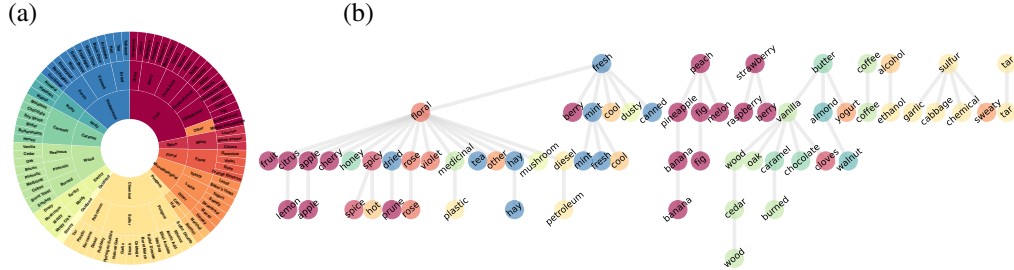

Figure 11: **LLM uncover wine aroma hierarchies aligning with the Davis Wine Aroma Wheel.** (a) Wine aroma wheel derived from Davis Wine Aroma Wheel[1]. (b) Hierarchical structure of wine aromas extracted from Llama 405B using 1,260 situational prompts generated by GPT-4. The tree was constructed using our algorithm based on logits from Llama 405B, revealing well-organized clusters that align with the categories in (a). This demonstrates the algorithm's ability to uncover meaningful hierarchical relationships solely from model representations, without relying on ground truth labels.

## C   DATA GENERATION AND MODELS FOR SECTION 3 AND 4

### C.1   COMPARING EMOTION HIERARCHY IN DIFFERENT MODELS

We construct a dataset by prompting GPT-4o (OpenAI, 2023) to generate 5000 sentences reflecting various emotional states, without specifying the emotion. We append the phrase "The emotion in this sentence is" after each sentence, before feeding it to the models we aim to extract emotion structures from. We extract the probability distribution over the next token predicted by the model, which represents the model's understanding of possible emotions for the given sentence. From the distribution of next token probabilities, we select the 100 most probable emotions for each sentence. We then construct the matching matrix as described in Section 3.1, and build the hierarchy tree.

To observe and compare the understanding of emotion hierarchy by different models, we construct the emotion trees using GPT2 (Radford et al., 2019), LLaMA 3.1 8B, LLaMA 3.1 70B, and LLaMA 3.1 405B (Dubey et al., 2024), with 1.5, 8, 70, and 405 billion parameters respectively. The Llama models are run using NNsight (Fiotto-Kaufman et al., 2024). We generate the tree layout using NetworkX (Hagberg et al., 2008).

### C.2   PROMPTS

#### C.2.1   GENERATING SCENARIOS USING GPT-4O

We use GPT-4o to generate scenarios without specifying the type of emotions with the following prompt:

```
Generate 5000 sentences.  Make the emotion expressed in the
sentences as diverse as possible.  The sentences may or may not
contain words that describe emotions.
```

To generate scenarios for specific emotions, we use the following prompts on GPT-4o, for each of the 135 emotion words. The first prompt generates stories from the third person view, without assuming the gender of the main character of the story. The second prompt generates stories from the first person view of a man or woman.

```
Generate 20 paragraph-long detailed description of different
scenarios that involves [emotion].  Each description must
include at least 4 sentences.  You may not use the word
describing [emotion].
```

```
Write 20 detailed stories about a [man/woman] feeling [emotion]
with the first person view.  Each story must be different.
Each story must include at least 4 sentences.  You may not use
the word describing [emotion].
```

#### C.2.2   EXTRACTING EMOTION USING LLAMA 405B

We ask Llama 3.1 405B to identify the emotion involved in a given scenario using the next word prediction on the following prompts. When not assuming any demographic categories, the prompt is *emotion scenario* + "The emotion in this sentence is". When assuming specific demographic groups, we use the prompts listed in Table 1.

## D   ADDITIONAL RESULTS

Figure 16 presents the hierarchical clustering results of internal representations for four models: (a) GPT-2 (1.5B parameters), (b) Llama-8B, (c) Llama 3.1-70B, and (d) Llama-405B. The x-axis displays emotion labels, color-coded by groups of related emotions. As model size increases, the emergence of deeper hierarchies reflects a finer-grained differentiation of emotions, consistent with our findings in Section 3. Notably, the emotion groupings produced by the LLMs diverge from

Table 1: Prompts used for extracting emotion predicted by Llama 3.1 405B.

| Categories | Prompt (*Emotion scenario* + _ + "I think ... ") |
|---|---|
| Gender | "As a [man/woman], " |
| Intersectional identities | 'As a [Black woman/low-income Black woman], " |
| Religion | "As a [Christian/Muslim/Buddhist/Hindu], " |
| Socioeconomic status | "As a [high/low]-income person, " |
| Age | "As a [5/10/20/30/70]-year-old, " |
| Ethnicity | "As a [White/Black/Hispanic/Asian] person, " |
| Education level | "As someone with [a postgraduate degree/a college degree/some college education/a high school diploma], " |
| Mental health | "As a person [with Autism Spectrum Disorder/experiencing depression/living with an anxiety disorder], " |
| Physical ability | "As [an able-bodied/a physically disabled] person, " |
| Detailed profiles | "As a [high-income/low-income] [White/Black] [man/woman], " |

(a) Llama 70B distilled variants of DeepSeek-R1[2]

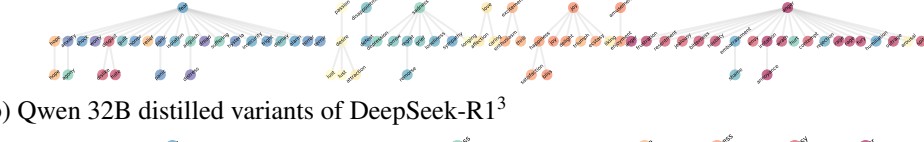

(b) Qwen 32B distilled variants of DeepSeek-R1[3]

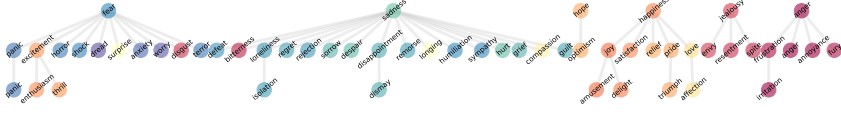

Figure 12: Hierarchies of emotions in two variants of DeepSeek-R1 models are extracted using 5000 situational prompts generated by GPT-4o. Each node represents an emotion and is colored according to groups of emotions known to be related (the emotion wheel in Fig. 1a).

established psychological frameworks. This contrast underscores the advantages of our proposed emotion tree (Figure 3) in providing a more accurate and comprehensive evaluation of LLMs' understanding of emotions.

Figure 17 shows the distance metrics of the emotion hierarchy: (a) total path length and (b) average depth, across different thresholds. Total path length captures the overall complexity of the hierarchy by summing all paths from the root to each leaf node, while average depth reflects how deep the hierarchy extends by calculating the mean distance from the root to the leaves. Similar to the trends seen in Figure 4, both metrics increase as model size grows. This suggests that larger models build more detailed and nuanced emotional hierarchies, improving their ability to represent the complexity of emotions.

Figure 18 compares the hierarchical emotion trees from Figure 3 with the human-annotated emotion wheel in Figure 1. To assess their relationships, clusters were extracted from the hierarchical emotion trees, and pairwise distances between emotions were defined based on cluster membership (0 if in the same cluster, 1 if in different clusters). We calculated the correlations between cluster distances and the color gaps on the emotion wheel, obtaining significant results: 0.55 for Llama-8B, 0.73 for Llama-70B, and 0.47 for Llama-405B, all with $p < 0.001$. These findings confirm the accuracy of the emotion structures derived from the LLMs. Additionally, we examined the relationship between the average number of hops between all pairs of nodes in the hierarchical trees and their corresponding distances on the emotion wheel. We see significant correlations: 0.55 for Llama-8B, 0.60 for Llama-70B, and 0.55 for Llama-405B, all at $p < 0.001$. These results further validate the reliability of the hierarchical emotion structures produced by the models.

In Figure 19, we present emotion wheels constructed from the hierarchical emotion trees in Figure 3 for (b) Llama-8B, (c) Llama-70B, and (d) Llama-405B, compared with (a) the original emotion wheel from psychological literature (Shaver et al., 1987), which is widely used in cognitive sci-

(a) Llama 8B

(b) Llama 70B

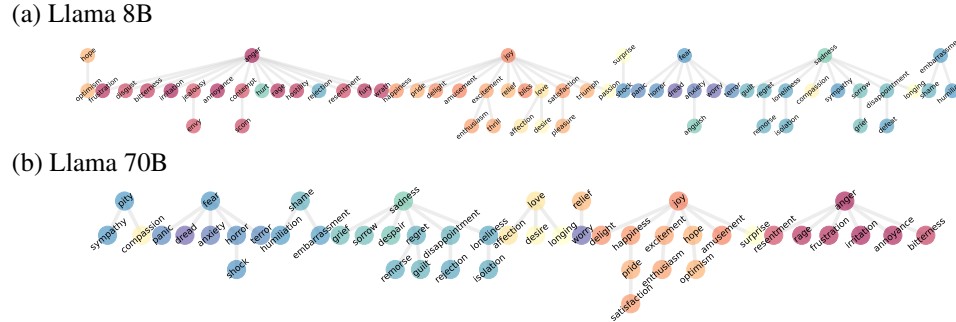

Figure 13: Sensitivity to prompting in hierarchical emotion clustering. We compare emotion trees produced when the instruction is changed from "the emotion in this sentence is" to "the sentiment in this sentence is." Across both models, similar emotions consistently cluster together, indicating robustness to prompt wording. The larger model, Llama 70B (b), produces deeper and more fine-grained hierarchies than the smaller Llama 8B (a), reflecting its greater representational capacity.

(a) Llama 8B

(b) Llama 70B

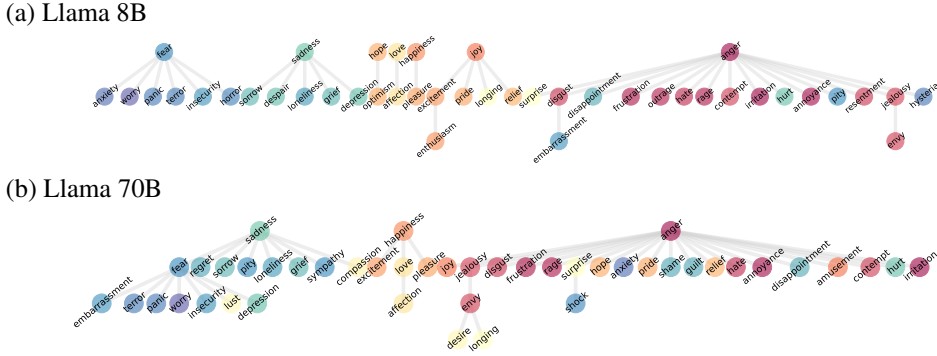

Figure 14: Hierarchies of emotions in Llama models using the GoEmotions dataset. Each node corresponds to an emotion and is colored according to groups of related emotions (as defined by the emotion wheel in Fig. 1a). The larger model, Llama 70B (b), produces deeper and more fine-grained hierarchies than the smaller Llama 8B (a), reflecting its greater representational capacity.

ence. We again observe that larger LLMs exhibit more hierarchical structures in their emotion trees. Moreover, the clustering in the larger models, (c) Llama-70B and (d) Llama-405B, shows greater alignment with the categories in (a) the original emotion wheel, compared to the smaller model, (b) Llama-8B.

Table 2: Difference in the predicted emotions and hierarchy for each pair of demographic groups.

| Demographic groups | # different predictions | # different edges in hierarchy |
|---|---|---|
| Gender (male/female) | 419 | 12 |
| Ethnicity (American/Asian) | 531 | 29 |
| Physical ability (able-bodied/disabled) | 744 | 43 |
| Socioeconomic (high/low income) | 707 | 36 |
| Education level (higher/less educated) | 400 | 27 |
| Age (10/30 years old) | 759 | 60 |
| Age (10/70 years old) | 798 | 69 |
| Age (30/70 years old) | 312 | 15 |

Figure 20 shows the difference between confusion matrices for various personas. Table 3 summarizes the observations in these confusion matrices. Table 2 shows the number of predictions (out of $135 \times 20 = 2700$) that Llama with each pair of persona (demographic groups) disagree. The table also quantifies the difference between the hierarchies generated from the prediction of each pair of demographic groups, by counting the number of different edges in the trees. We generate the

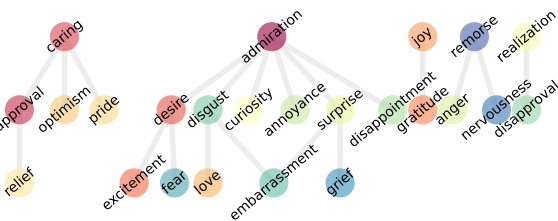

Figure 15: The emotion tree was constructed from the GoEmotions dataset using multi-label annotations rather than an LLM's logits. The resulting tree is smaller than the one derived from LLM logits because human annotators provide only one or a few labels per instance, which limits label co-occurrence and leads to a more compact hierarchical structure.

Table 3: Difference in the predictions by each pair of different demographic groups, obtained by comparing confusion matrices.

| Demographic A | Demographic B | More often predicted by A | More often predicted by B |
|---|---|---|---|
| Male | Female | - | jealousy |
| Asian | American | shame | embarrassment |
| Able-bodied | Disabled | excitement, anxiety | hope, frustration, loneliness |
| High income | Low income | excitement | happiness, hope, frustration |
| Highly educated | Less educated | grief, disappointment, anxiety | happiness |
| Age 30 | Age 10 | frustration | happiness, excitement |
| Age 70 | Age 30 | loneliness | excitement, frustration |

hierarchies using the method described in Section 3.1, with threshold 0.3. Most trees have around 100 edges.

Figure 23 shows emotion recognition accuracy across six broad emotion categories for human participants in the user study. Comparing this with Figure 5 highlights notable differences: (a) human females outperform males, while Llama shows the opposite trend, favoring males. Llama also mirrors human biases across (b) race and (c) education levels, with Black and White participants performing worse than Hispanic and Asian participants, and higher education levels correlating with better performance.

Figure 24 shows Llama's misclassification patterns, highlighting intersectional biases across demographic groups. The chord diagram in this figure visually represents the flow of misclassified emotions between emotion categories for four demographic groups: (a) high-income Black males, (b) White individuals, (c) low-income White females, and (d) low-income Black females. In panel (b), high-income Black males exhibit a notable misclassification of fear as anger, whereas in panel (a), White individuals display fewer such errors. Panel (c) shows that low-income White females tend to misclassify emotions as fear. In contrast, panel (d) demonstrates that low-income Black females exhibit a combination of these biases, resulting in lower overall accuracy. This analysis further highlights the amplification of LLM's emotion recognition biases for intersectional underrepresented groups, where misclassifications are more pronounced, impacting both model performance and fairness.

Table 4 shows the accuracy and F1 scores of Llama 405B across personas that combine income, race, and gender for emotion recognition on 2700 situational prompts representing broad six emotions generated by GPT-4o. We observe that the low-income Black female persona exhibits the lowest performance on both metrics, indicating a pronounced intersectional bias in the LLM's emotion recognition. In contrast, high-income Black personas achieve the highest accuracy and F1 scores. This pattern suggests that adding positive socioeconomic attributes such as high income may improve performance in ways that go beyond mitigating racial bias. Understanding why these positive modifiers help could be an interesting direction for future work.

Figure 25 compares how the emotion "surprise" is misclassified into other emotions by Llama 40B (top) and humans (bottom). For humans, the neutral persona condition represents the average perfor-

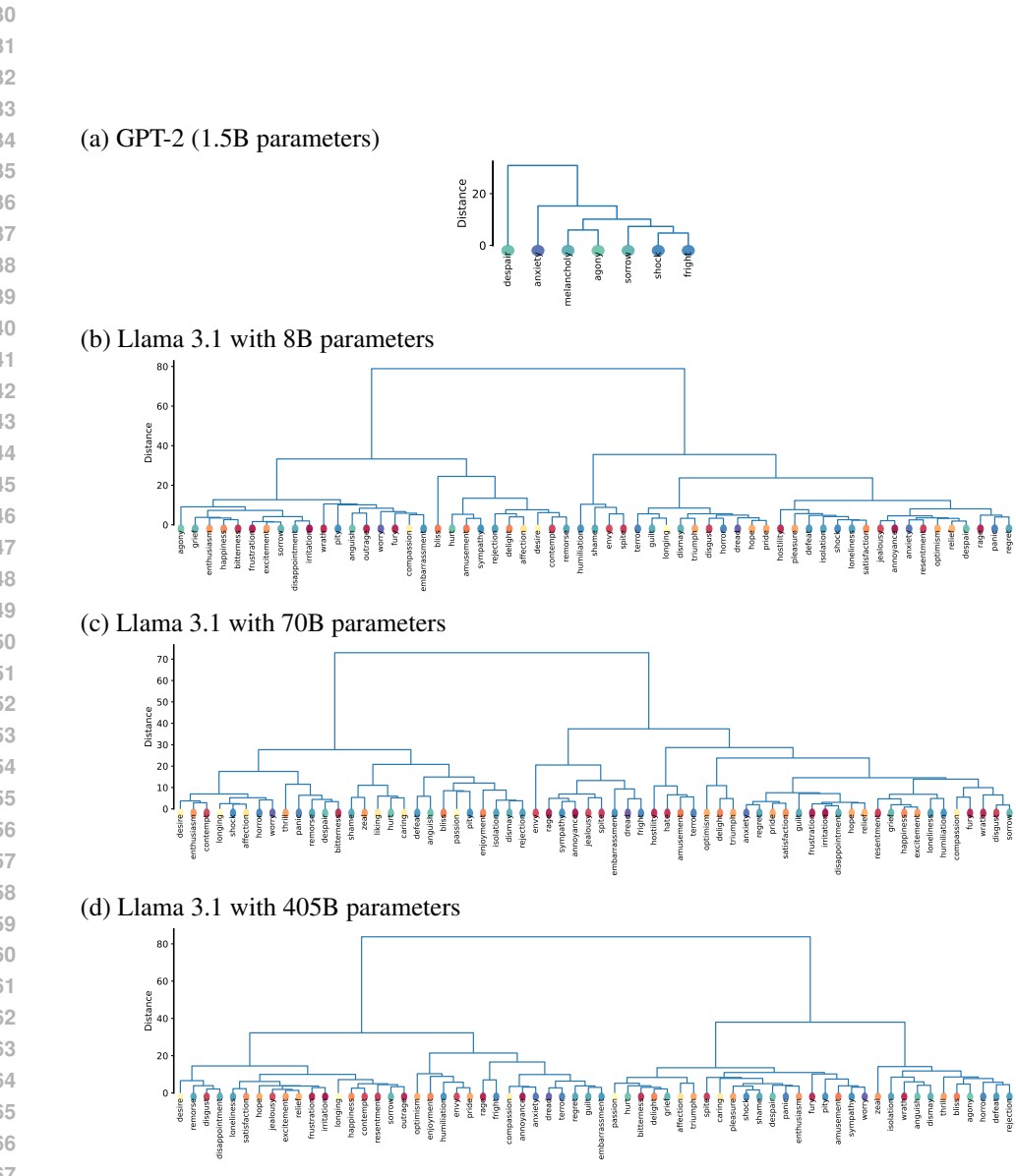

Figure 16: Hierarchical clustering of internal representations for 135 emotions, derived from four models: (a) GPT-2 (1.5B parameters), (b) Llama-8B, (c) Llama 3.1-70B, and (d) Llama-405B, using 5,000 situational prompts generated by GPT-4o. As model size increases, more hierarchies emerge, reflecting finer-grained differentiation of emotions. Each node represents an emotion and is colored according to groups of emotions known to be related.

(a) (b)

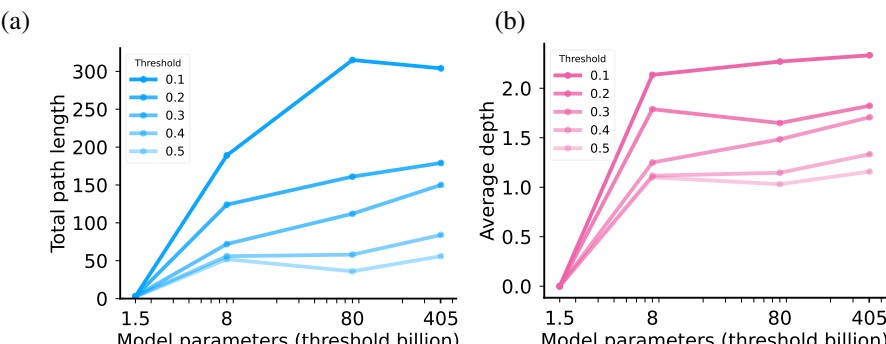

Figure 17: The distance metrics of the emotion hierarchy, (a) total path length and (b) average depth, are plotted as functions of model size across various thresholds. We see robust trend across different threshold selections: as model size increases, both measures grow, suggesting that larger models construct more complex and nuanced emotional hierarchies.

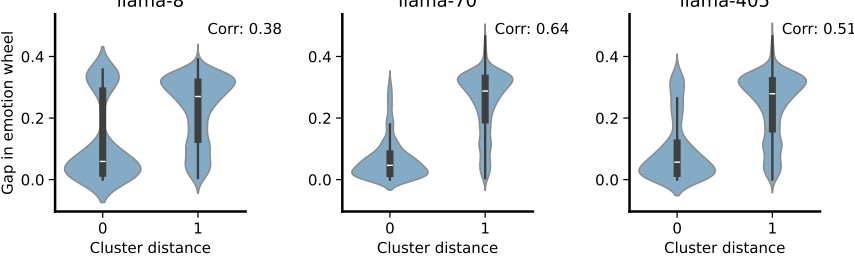

Figure 18: **Hierarchical emotion structures derived from Llama models align closely with human-annotated emotion relationships.** Quantitative comparison of hierarchical emotion trees from Llama models (8B, 70B, and 405B) with the human-annotated emotion wheel. (a) Correlations between cluster distances in the hierarchical trees and color gaps on the emotion wheel show significant alignment ($p < 0.001$), demonstrating the accuracy of the LLM-derived emotion structures. (b) Correlations between node hops in the hierarchical trees and corresponding distances on the emotion wheel further validate the integrity of the extracted emotion hierarchies, with all results significant at $p < 0.001$.

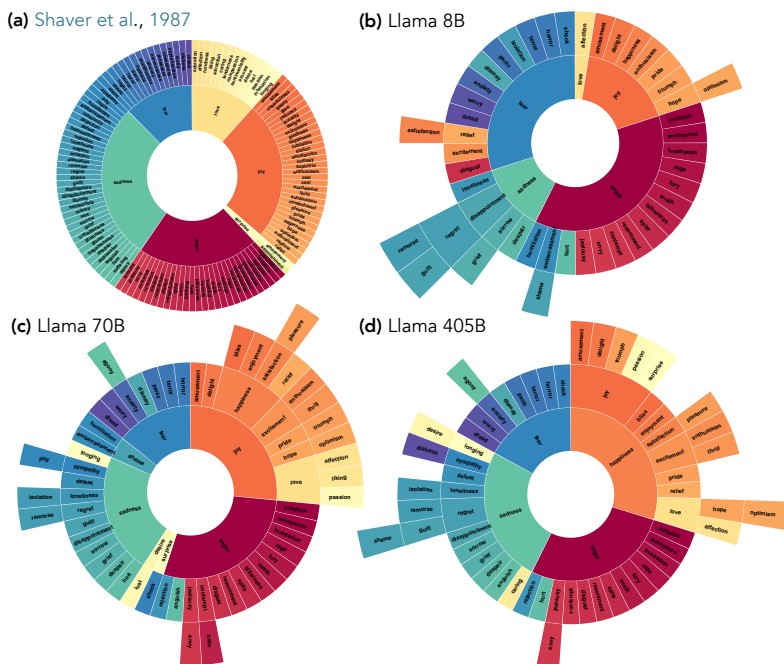

Figure 19: **Larger LLMs construct emotion wheels with deeper hierarchies and better-aligned groupings.** (a) The original emotion wheel from psychological literature (Shaver et al., 1987). Hierarchical emotion trees constructed for (b) Llama-8B, (c) Llama-70B, and (d) Llama-405B. As the model size increases, the trees exhibit deeper and more refined hierarchical structures, demonstrating the enhanced capacity of larger models to represent complex relationships between emotions.

Table 4: Emotion recognition accuracy and F1 score of Llama 405B on 2700 situational emotion recognition prompts for broad six emotions, evaluated across personas varying in income, race, and gender.

| LLM Persona | Accuracy | F1 |
|---|---|---|
| High income White male | $0.804 \pm 0.020$ | 0.793 |
| High income White female | $0.795 \pm 0.020$ | 0.812 |
| High income Black male | $0.814 \pm 0.015$ | 0.806 |
| High income Black female | $0.805 \pm 0.017$ | 0.798 |
| Low income White male | $0.798 \pm 0.017$ | 0.759 |
| Low income White female | $0.759 \pm 0.026$ | 0.771 |
| Low income Black male | $0.771 \pm 0.056$ | 0.767 |
| Low income Black female | $0.739 \pm 0.064$ | 0.736 |

mance of 60 participants in the user study. In this condition, Llama misclassifies "surprise" mainly as "fear", achieving an accuracy of 41.7% compared to 56.4% for humans. Llama's accuracy declines further when adopting personas, particularly for underrepresented groups. For instance, it correctly identifies "surprise" only 17.2% of the time for females and 6.7% for Black individuals, whereas human performance remains more consistent across demographics. This highlights Llama's biases, which differ from natural human tendencies and should be addressed.

In Figure 26, we construct hierarchical emotion trees from Llama 405B logits, using different personas as described in Section 4, following the methodology in Section 3. The hierarchical structures become more complex for personas with higher emotion recognition accuracy. (a) high-income white male has higher emotion prediction accuracy show the most complex structures, with a larger number of nodes, especially in the second and third layers. (b) The high-income white female and (c) low-income black female personas have moderately lower accuracy and simpler structures. (d) Physically-disabled personas show the simplest structures, with significantly fewer nodes in the

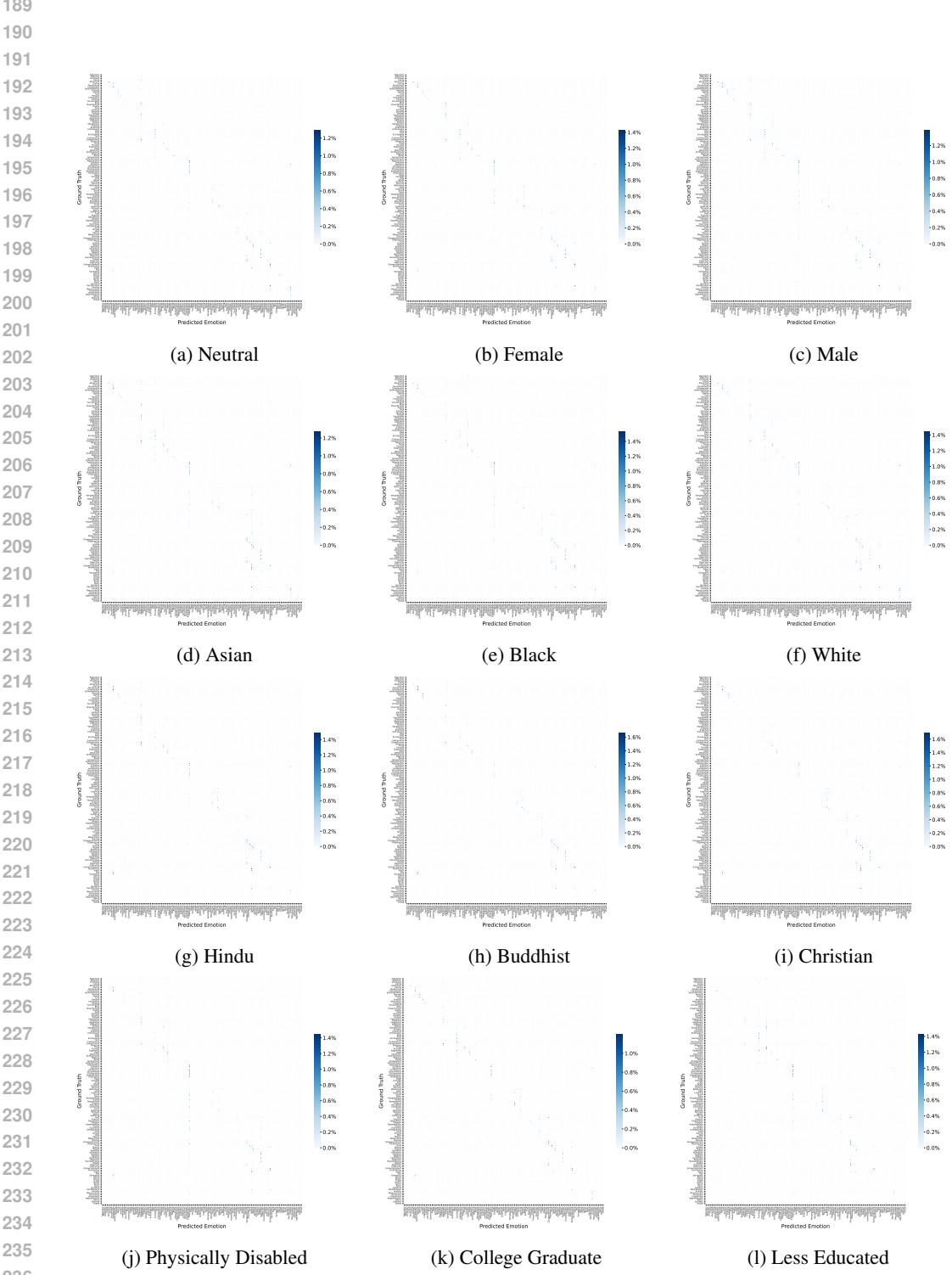

(a) Neutral

(b) Female

(c) Male

(d) Asian

(e) Black

(f) White

(g) Hindu

(h) Buddhist

(i) Christian

(j) Physically Disabled

(k) College Graduate

(l) Less Educated

Figure 20: Confusion matrices show the performance of different personas in recognizing 135 distinct emotions, highlighting variations in emotion perception and classification accuracy.

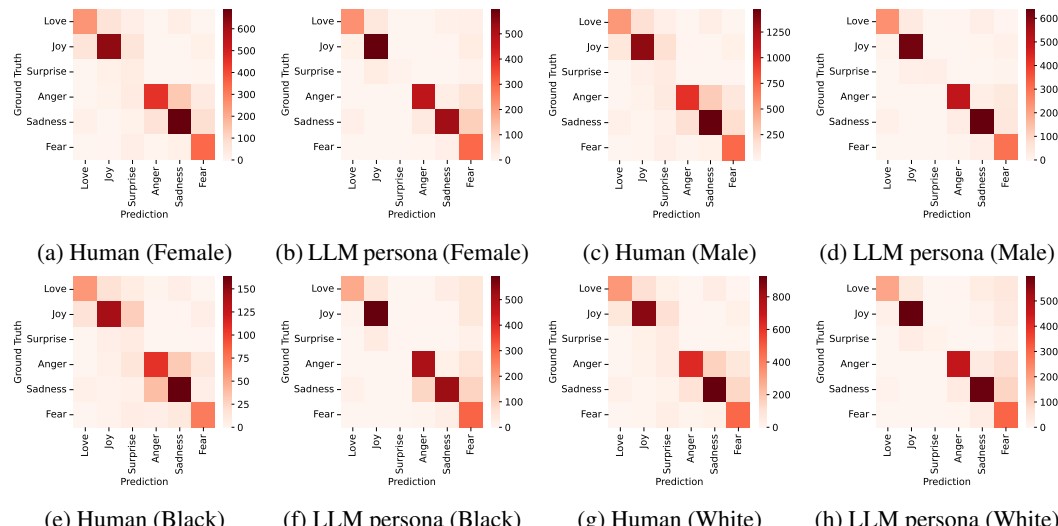

(a) Human (Female)    (b) LLM persona (Female)    (c) Human (Male)    (d) LLM persona (Male)

(e) Human (Black)    (f) LLM persona (Black)    (g) Human (White)    (h) LLM persona (White)

Figure 21: Confusion matrices for human participants and LLM personas across gender (female, male) and racial (Black, White) identities.

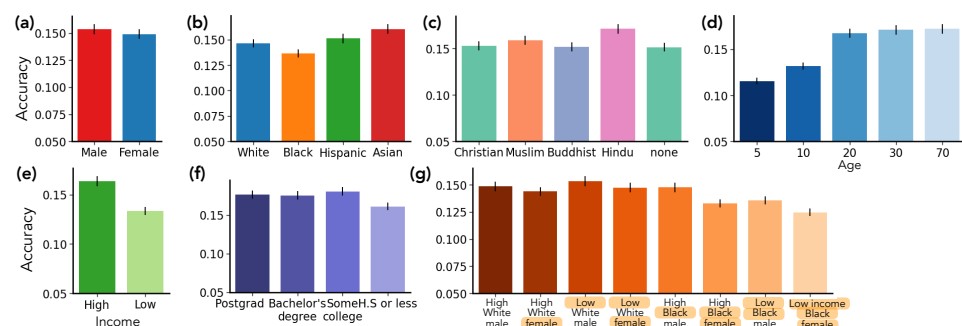

Figure 22: **LLM has lower accuracy in emotion recognition for underrepresented groups compared to majority groups.** We assessed the model's performance in predicting 135 emotions across demographic group. Llama 405B consistently struggles to accurately recognize emotions in underrepresented groups, such as (a) females, (b) Black personas, (e) individuals with low income, and (f) individuals with low education, compared to majority groups. These performance gaps are even more pronounced when multiple minority attributes are combined (g), such as in the case of low-income Black females.

lower layers and the lowet emotion recognition accuracy. This gradation suggests the hierarchical emotion tree reflects the LLM's intrinsic emotional understanding, which directly impacts emotion recognition accuracy.

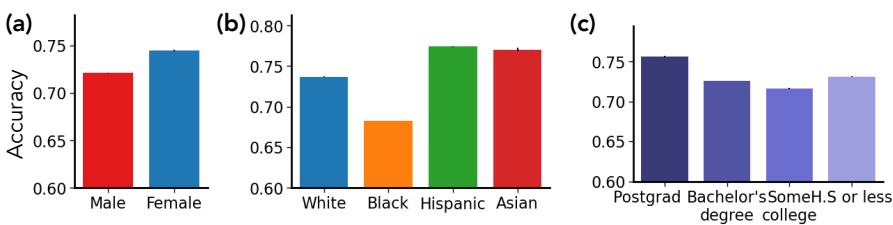

Figure 23: **Human biases align with LLMs across race and education, but the gender bias is reversed, with humans favoring females and LLMs favoring males.** Emotion recognition accuracy for six broad emotion categories among human participants in the user study. Comparison with Figure 5 highlights notable differences between LLM and human performance: (a) human females outperform males, while Llama exhibits a reversed bias, favoring males. Additionally, Llama replicates human biases in emotion classification, with (b) Black and White participants performing worse than Hispanic and Asian participants, and (c) higher education levels correlating with better emotion recognition accuracy.

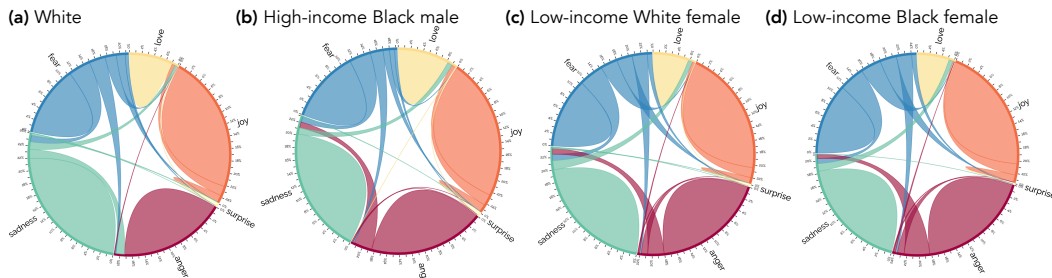

Figure 24: **LLM's emotion recognition biases are amplified for intersectional underrepresented groups.** Llama's misclassification patterns reveal intersectional biases across demographic groups. (b) high-income black males often misclassify fear as anger, (a) White personas show fewer such errors, (c) low-income white females tend to misclassify emotions as fear, and (d) low-income black females combine these biases, leading to lower accuracy.

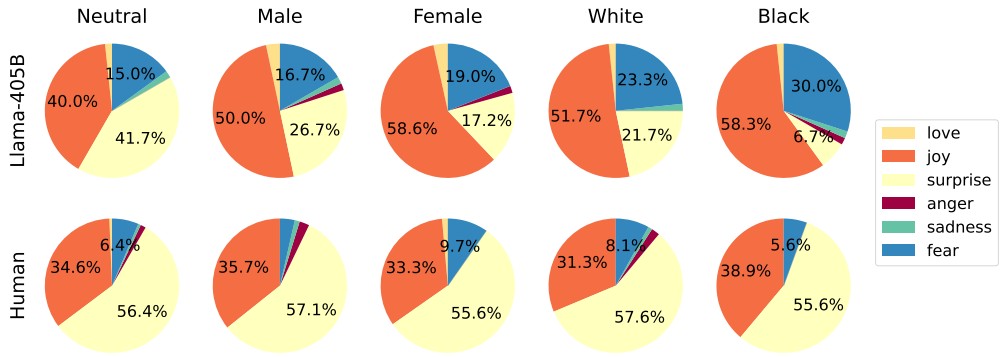

Figure 25: **LLMs struggle more with accurately recognizing emotions compared to humans.** Comparison of emotion "surprise" misclassification patterns between Llama 40B (top) and humans (bottom). In the neutral persona condition, Llama misclassifies "surprise" primarily as "fear", with an accuracy rate of 41.7% compared to 56.4% for humans. When adopting personas, Llama's accuracy drops significantly, especially for underrepresented groups such as female (17.2%) and Black personas (6.7%), whereas human performance remains more consistent across demographics.

(a) High-income white male persona by Llama 405B

(b) High-income white female persona by Llama 405B

(c) Low-income black female persona by Llama 405B

(d) Physically-disabled persona by Llama 405B

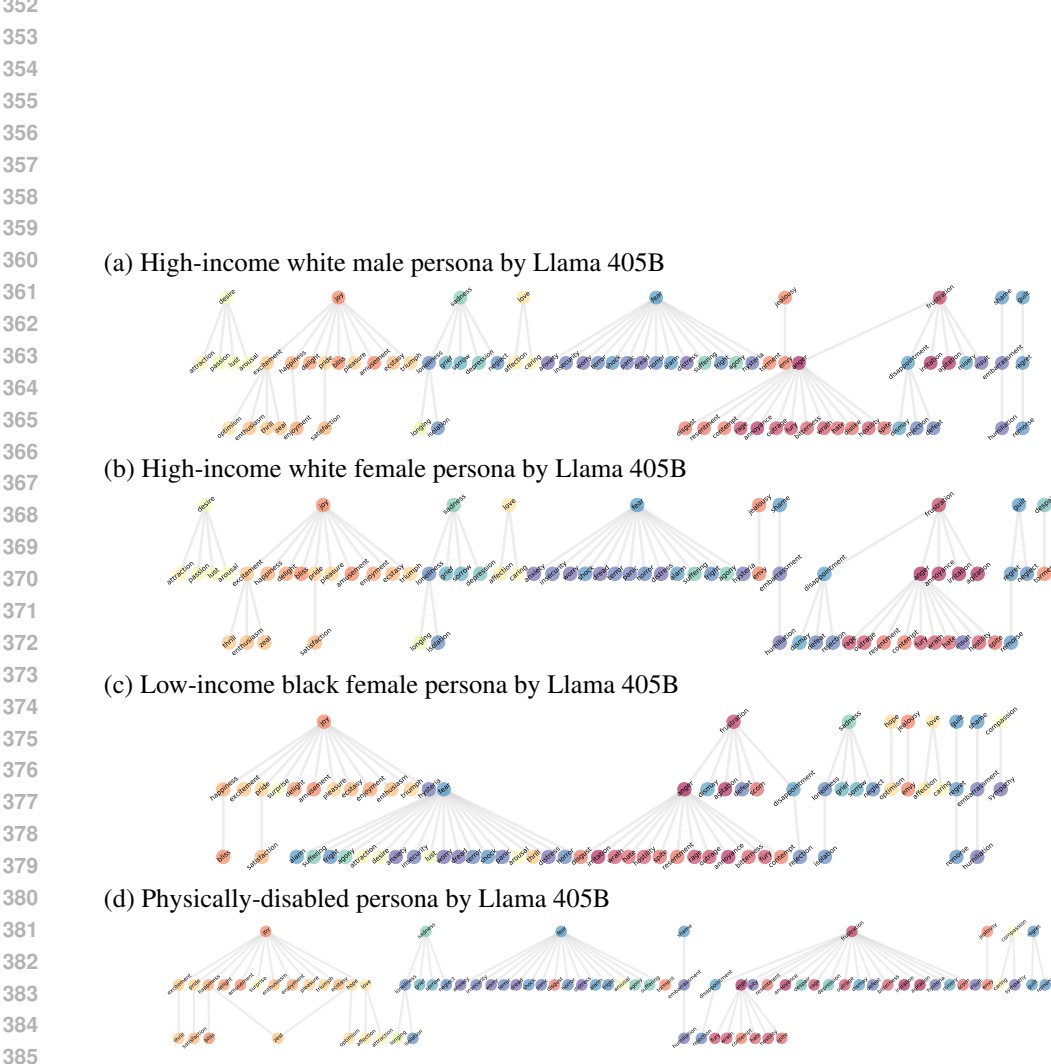

Figure 26: Hierarchies of emotions in Llama 405B across different personas, extracted using 2,700 situational prompts for 135 emotions generated by GPT-4o. Each node represents an emotion, colored by related emotion groups (shown in the emotion wheel, Figure 1). (a) The high-income white male persona shows the most complex structure, with a larger number of nodes in the second and third layers, corresponding to higher emotion recognition accuracy. (b) The high-income white female and (c) low-income black female personas exhibit moderately simpler structures and lower accuracy. (d) The physically-disabled persona has the simplest structure, with fewer nodes in the lower layers and the lowest recognition accuracy. This suggests that the emotion tree reflects LLMs' intrinsic emotional understanding, which impacts the accuracy of emotion recognition.

# E EXPANDING TO HUMAN-ANNOTATED DATA

We compare human annotations and Llama's predictions using the GoEmotions dataset (Demszky et al., 2020). Figure 27 shows mismatches between human labels and Llama's outputs across 27 emotion categories. Notably, when using (c) a female persona, Llama often misclassifies various emotions as fear, and when using (d) a Black persona, it tends to misclassify them as anger—trends that are consistent with our earlier findings.

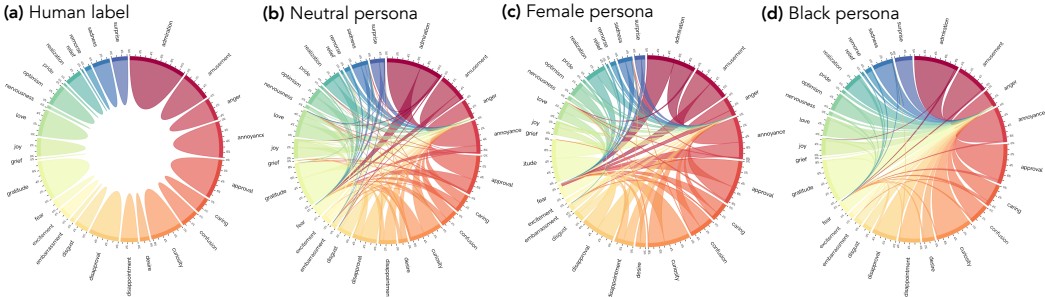

Figure 27: **LLMs demonstrate consistent biases in emotion recognition with underrepresented groups.** We used the GoEmotions dataset (Demszky et al., 2020) to compare Llama's emotion recognition performance against human-labeled data across 27 emotion categories. Llama shows consistent biases, frequently misclassifying emotions as fear for female personas (c) and as anger for Black personas (d), compared to neutral persona (b).

