# OpenReview forum: "Emergence of Hierarchical Emotion Organization in Large Language Models"
_ICLR.cc/2026/Conference — Submitted to ICLR 2026_

### Official Review · Reviewer_oZ7X · 2025-10-26

**Soundness:** 3
**Presentation:** 3
**Contribution:** 2
**Rating:** 4
**Confidence:** 3

**Summary:**

This paper investigates the internal organization of emotional understanding within LLMs, drawing inspiration from human psychological models like the emotion wheel.
The authors propose a novel algorithm to extract a hierarchical "emotion tree" based on probabilistic dependencies between emotional states predicted by the LLMs.
Experiments on the Llama family of models show that these hierarchies evolve with model size and are consistent with human psychology.
Additionally, the emotional recognition capabilities are prone to systematic biases across different socioeconomic personas.
Overall, the work contributes a complementary perspective to standard emotion classification benchmarks, emphasizing emergent structures in LLMs.

**Strengths:**

1. The paper is clearly structured and and the analysis is interesting.
2. The tree-construction algorithm is innovative and easily extendable to other classification tasks (such as wine aroma hierarchy).

**Weaknesses:**

1. The description of Exp1 and Exp2 in Figure 2 is not clear enough, especially in the top right part. For the readers' ease of understanding and readability, the reviewer suggests that the location of Exp1 should be explicitly marked.
2. The data sources lack diversity. Experiments 1 and 2 used synthetic data generated by GPT-4o, resulting in an overly homogenous data distribution. Furthermore, the paper also notes that LLM has biases in understanding certain emotions (for example, difficulty identifying "surprise"), which may lead to bias in the generated data.
3. The paper mainly conducts experiments on the LLaMA series of models, which is too single. It lacks experiments on more types of models (such as large reasoning models and MoE architecture models).
4. In the Discussion section, the authors claim "reinforcement learning helps close this gap by rewarding reductions in prediction error," but do not explain it.

Minor comments:
- The subtitles (a,b,c,d) and caption in Figure 8 are inconsistent with lines 370-372.

**Questions:**

1. Experiment 1 extracts the probability distribution of the next token by appending "The emotion in this sentence is" to each prompt. However, LLM is very sensitive to prompts. Have you conducted an experiment to verify the robustness of this extraction method using different prompts (e.g., "The emotion in this text is" / "The emotion is" / "The sentiment is").
2. For Experiment 1, due to the inherent bias of LLMs, did the authors count the frequency of each emotion? Would this affect the experimental conclusions?

---

> ### Author Response · Authors · 2025-12-03
>
> We thank the reviewer for the constructive and detailed feedback. We added **four new experiments** for you and revised the manuscript accordingly. We believe the additions covering MoE models, alternative prompts, RL-fine-tuned models, and human-annotated GoEmotions data address **all** raised concerns. Below we provide a point-by-point response.
>
> ---
>
> **Weakness  1. Clarity of Experiment 1 & 2 in Figure 2**
> > The description of Exp1 and Exp2 in Figure 2 is not clear enough… the location of Exp1 should be explicitly marked.
>
> Thank you for pointing this out. We have updated **Fig. 2** to clearly mark the connection between Exp 1 and Exp2.
>
> ---
>
> **Weakness 2. Data source diversity & potential bias in synthetic data**
> > Experiments 1 and 2 rely on synthetic data generated by GPT-4o… this may introduce emotion-recognition biases (e.g., difficulty with “surprise”).
>
> We appreciate the reviewer's concern. In the revised version, we added new experiments using **human-annotated GoEmotions data in Fig. 14**. The results are consistent with our earlier observations:
> - The learned emotion categories still closely align with the Shaver emotion wheel.
> - Larger models continue to yield deeper hierarchies,
> which shows robustness of our tree construction algorithm.
>
> ---
>
> **Weakness 3. Limited model diversity (LLaMA-centric experiments)**
> > The paper mainly focuses on LLaMA models… lacks experiments on other architectures such as reasoning models or MoE models.
>
> Thank you for this helpful suggestion. We expanded our analysis beyond the Llama family by adding a new experiment using **MoE-based DeepSeek-distilled models** (see **Fig. 12**).
>
> Key findings:
> - These MoE models produce emotion clusters that closely follow the psychology-grounded emotion wheel.
> - Compared to Llama, the MoE variants yield **slightly fewer** distinct emotion clusters.
> - A plausible explanation is that MoE models often generate more convergent, less diverse outputs (Dang et al., 2025).
>
> Experiments on larger DeepSeek models are ongoing and will be included in the camera-ready version.
>
> (Dang et al., 2025) Dang, Xingyu, et al. "Assessing diversity collapse in reasoning." Scaling Self-Improving Foundation Models without Human Supervision, 2025.
>
> ---
>
> **Weakness 4. Unexplained claim about reinforcement learning improving prediction error**
> > The Discussion section claims that RL “helps close this gap,” but this is not explained.
>
> We thank the reviewer for pointing this out. We added a new subsection in **Section 4.2** and a new **Fig. 10**, comparing the base Mistral-7B with its RL-fine-tuned version. Accuracy on **surprise** increases from **20.0% → 33.3%**, a statistically significant improvement. This provides empirical support that RL can mitigate specific failures in emotion recognition.
>
> ---
>
> **Minor comment: Inconsistency in Figure 8 caption/subtitles**
> > The subtitles (a, b, c, d) and caption in Figure 8 are inconsistent.
>
> Thank you for catching this. We have corrected the subtitles and caption in **Fig. 8**.
>
> ---
>
> **Question 1. Prompt sensitivity in Experiment 1**
>
> > LLMs are sensitive to prompt wording. Did you test prompts such as “The emotion is…” or “The sentiment is…”?
>
> We added new experiments with alternative prompt formulations (see **Fig. 13**). We found
> - “The emotion is…” and “The sentiment is…” yield nearly identical hierarchical clusters.
> - Larger models still produce deeper emotion trees.
>
> This shows our extraction method is robust to prompt variations. More prompt variations and larger models (e.g., Llama 405B) will be included in the camera-ready version.
>
> ---
>
> **Q2. Emotion frequency bias and its effect on Experiment 1**
>
> > Did the authors count the frequency of each emotion? Could frequency bias affect conclusions?
>
> To address this, we conducted new experiments using **GoEmotions**, which has a label distribution very different from GPT-4o-generated data (see **Fig. 15**). Despite the distribution differences, the hierarchical trees derived from GoEmotions (**Fig. 14**) remain qualitatively consistent with earlier results.
> This suggests that our methodology is robust to underlying emotion frequency variations.

---

### Official Review · Reviewer_ij2x · 2025-10-30

**Soundness:** 3
**Presentation:** 3
**Contribution:** 3
**Rating:** 6
**Confidence:** 3

**Summary:**

This paper introduces a novel benchmark to evaluate large language models’ (LLMs) understanding of human emotions.
Inspired by psychological theory, the authors design a framework based on the emotion wheel to analyze how LLMs recognize emotions hierarchically.
They further examine how these models behave under different personas (e.g., high-income vs. low-income, gender, and race) to uncover systematic demographic biases in emotion recognition.
The study finds that larger models tend to form more structured hierarchical emotion representations, while also revealing fairness issues when simulating underrepresented groups.

**Strengths:**

- Novelty: The paper creatively uses the emotion wheel—a psychological model—to evaluate hierarchical emotion recognition in LLMs.

- Relevance: As LLMs increasingly engage in human-like dialogue, studying their emotional recognition and biases is timely and practically valuable.

- Clarity: The presentation is clear and easy to follow. The logical flow helps readers understand both the method and the experiment design.

- Comprehensive analysis: The authors test multiple settings, including different model sizes and assigned personas, to evaluate bias in emotion recognition, which provides valuable insight in practice.

**Weaknesses:**

- Some related works, such as [1], are neglected.

- The analysis of demographic bias (e.g., Figure 7) only covers two emotions (“anger” and “fear”). It would be better to include overall accuracy or more emotion categories to show whether the overall bias is consistent.

- The evaluation data are primarily generated by GPT-4, raising questions about whether the performance and conclusions remain valid for real data produced by humans.

- The hierarchical emotion relations are derived solely from model logits, which appears to be an indirect approach. It seems there is a lack of assessment for the validity of the results.



[1] Clever Hans or Neural Theory of Mind? Stress Testing Social Reasoning in Large Language Models. EACL 2024

**Questions:**

- The authors define the model’s bias as the variation in emotion recognition accuracy when the model is assigned different personas. However, I would suggest considering an alternative perspective: the model could be viewed as an objective evaluator, and bias might more appropriately refer to how its predictions change when recognizing inputs of users from different demographic groups. For instance, the model could be prompted with: "This sentence was written by a low-income White male. What emotion does it express?" Such an approach might better capture practical, real-world aspects of bias.



- Line 419 typo: Only left parentheses, no right parentheses.

---

> ### Author Response · Authors · 2025-12-03
>
> We thank the reviewer for the constructive feedback. In response to your comment, we added several concrete analyses, including new experiments using human-annotated data (Fig. 14), overall accuracy across all six emotions (Table 4), expanded validation of our logit-based hierarchy, and clarifications on key modeling assumptions and bias framing, which we believe fully address all of your concerns. Below we address each point one by one.
>
> ---
>
> **Weakness 1. Missing related work (e.g., [1])**
> > Some related works, such as [1], are neglected.
>
> Thank you for the pointer to [1], which we find very interesting. While [1] stress-tests social reasoning and theory-of-mind abilities using explicit social-reasoning tasks and focuses on distinguishing genuine reasoning from “Clever Hans” artifacts; our work is complementary:
> (i) we study how hierarchical emotion structure emerges **directly** in model logits, and
> (ii) we connect this internal organization to systematic emotion-recognition biases across demographic personas and their fairness implications.
> In the revised manuscript, we now explicitly cite [1] in the **Related Work** section and discuss how our study relates to and differs from theirs.
>
> ---
>
> **Weakness 2. Limited emotion coverage in demographic-bias analysis**
> > The analysis of demographic bias (e.g., Figure 7) only covers two emotions (“anger” and “fear”). It would be better to include overall accuracy or more emotion categories to show whether the overall bias is consistent.
>
> Motivated by this suggestion, we now report **overall emotion-classification accuracy across all six categories**, using 2,700 situational descriptions. The results show that intersectional minority personas exhibit the lowest accuracy (e.g., **0.739 ± 0.064** for a low-income Black female persona).
>
> This demonstrates that the biases we found are **not** limited to “anger” and “fear,” but instead reflect broader, consistent misclassification patterns across the full set of emotions.
>
> ---
>
> **Weakness 3. Human-generated vs. GPT-4-generated evaluation data**
> > The evaluation data are primarily generated by GPT-4, raising questions about whether the performance and conclusions remain valid for real data produced by humans.
>
> We added new experiments in **Fig. 14**, presenting emotion trees for Llama models constructed from **human-annotated GoEmotions** data. The results are consistent with our earlier observations:
> - the resulting clusters align well with the Shaver emotion wheel, and
> - larger Llama models produce deeper hierarchical trees.
>
> This shows that our conclusions remain stable even when trained on or evaluated using real human-annotated data.
>
> ---
>
> **Weakness 4. Validity of hierarchical relations derived from logits**
> > The hierarchical emotion relations are derived solely from model logits, which appears to be an indirect approach. It seems there is a lack of assessment for the validity of the results.
>
> We agree that deriving hierarchies from logits can appear indirect. To address this, we now make both the **probabilistic grounding** and **empirical validation** explicit. As shown in Appendix A, under the standard interpretation of next-token probabilities, our matching matrix and edge criteria estimate asymmetric conditional probabilities \( P(e_b \mid e_a) \). Thus, each edge encodes that a more specific emotion strongly implies a more general one.
>
> We further validate the resulting hierarchies in three ways:
>
> 1. **Alignment with psychology:**
> Emotion clusters and hop distances strongly correlate with Shaver wheel distances (up to \( r = 0.73 \), \( p < 0.001 \); Figs. 1, 15–16).
>
> 2. **Domain-general recovery of hierarchical structure:**
> The same method recovers meaningful hierarchies for **wine aroma descriptors**, matching the Davis Wine Aroma Wheel (Fig. 11).
>
> 3. **Predictive validity:**
> Geometric properties of the emotion trees (total path length, average depth) strongly predict persona-conditioned emotion-recognition accuracy across 26 personas (\( r = 0.84 \) and \( r = 0.48 \); Fig. 6).
>
> Finally, we contrast our method with hierarchical clustering based on **internal representations** (Fig. 16), which shows substantially weaker alignment with psychological structure. This further supports the validity and usefulness of our logits-based hierarchy construction.

---

> ### Author Response · Authors · 2025-12-03
>
> **Question 1. Alternative definition of bias focusing on the input author**
> > The authors define the model’s bias as the variation in emotion-recognition accuracy when different personas are assigned. However, one could alternatively view bias as how predictions change when the author's demographics vary (e.g., “This sentence was written by a low-income White male…”).
>
> Thank you for this insightful comment. We agree that analyzing **author demographic bias** is valuable and captures an important real-world perspective.
>
> Our current work focuses on a **complementary** dimension: **persona-induced bias**. This is crucial because persona prompting is known to expose deeply embedded reasoning biases (Gupta et al., 2023), and adjusting personas can meaningfully change LLM performance across diverse tasks (Ghandeharioun et al., 2024, Tan et al., 2025).
>
> We agree that the reviewer’s proposed direction is important, and we will incorporate author-demographic bias analyses as a key avenue for future work.
>
> (Gupta et al., 2023) Gupta et al., 2023. *Bias Runs Deep: Implicit Reasoning Biases in Persona-Assigned LLMs.*
> (Ghandeharioun et al., 2024) Ghandeharioun et al., 2024. *Who’s Asking? User Personas and the Mechanics of Latent Misalignment.*
> (Tan et al., 2025) Tan, Bryan et al., 2025. *Unmasking Implicit Bias: Evaluating Persona-Prompted LLM Responses in Power-Disparate Social Scenarios.*
>
> ---
>
> **Question 2: Typo**
> Thanks for catching it. We have corrected it.

---

### Official Review · Reviewer_VfGQ · 2025-10-31

**Soundness:** 1
**Presentation:** 1
**Contribution:** 1
**Rating:** 0
**Confidence:** 4

**Summary:**

This paper seeks to understand how LLMs  recognize and model emotions in the context of a conversation. It does this by constructing a hierarchy from the LLM logits. The authors never directly evaluate this method. The authors then explore model bias in how a persona classifies emotions.

**Strengths:**

The method from Section is a good method and an interesting way to extract emotional hierarchies from LLMs.

**Weaknesses:**

The paper presents its results with a lack of clarity and care for the reader. The methodological contribution is never properly evaluated against a ground truth. The text of the paper contradicts the results presented. This paper should and needs to be rewritten for it to be an interesting and meaningful contribution. My suggestion is to focus on the method from section 3 and flesh it out. Furthermore, if the author wants to claim model bias, they should (a) make sure that the stated bias aligns with their results and (b) explore why this bias exists. There are many papers on model biases, but without an exploration of how the bias in this case arises and how to potentially mitigate it (or land in a different area of the data distribution that may no longer have the specific bias), this particular exploration adds little new information.

The methodology of using a persona is also not well justified. LLMs do not experientially experience the world. They furthermore do not experientially experience being “White”, “Black”, “Hispanic”, “Asian”, “Low-Income”, “High-income”. At most this can show what these terms were associated with in the pretraining dataset. Is it possible that these models just do not see as much text relating to “low-income” persons or from “low-income” persons? If so, how can the authors claim that the issue is with the models and not the training dataset? Without treatment of this subject, it is not clear what conclusions the authors want the reader to takeaway with regard to reducing model bias.

-	Weaknesses:
o	There is no direct evaluation of the method presented in section 3. Why is the accuracy of the method not measured against a multi-label dataset?
o	Line 310-311 is directly contradicted by Figure 5. Neither Hispanic nor Asian is majority demographic and the Llama-405B persona for both of those outperform the persona for White. “Some college” outperforms all other categories. “Low White Male” outperforms “High White Male”, “Low White Female” performs about as well as “High White Male” and outperforms “Low White Female”, “High Black Male” outperforms or performs just as well as all White categories except for “Low White Male”. “Low Black Male”, “Low Black Female”, and “High Black Female” all underperform the other categories. The paper would be more interesting if a more nuanced view was taken instead of just taking the simplistic framing of “majority” vs “minority”, as that is not what seems to be happening in the results.
	Furthermore, in Figure 16 this pattern continues with both “Hindu” and “Muslim” outperforming “Christian” and “None” and “Buddhist” performing about the same.
	For age, the model performs poorly for ages 5 and 10, but I would guess that that is because there is less data online on the experience of being 5 and 10.
o	Figure 5c xlabels can be spaced apart a bit more.
o	Expand on lines 319-320. It was difficult to follow what Figure 6 was referring to. Not enough introduction.
o	Having a persona be “depression” seems to be flawed as depression is directly related to being able to only experience a certain set of emotions and with emotional listlessness. I am not surprised that a “depressed” persona is less able to recognize emotions.
o	Figure 6, further refutes the simplistic framing this paper chose with “majority” vs “minority”.
o	Figure 7 seems to be cherry-picked. If you want to show misclassification error, use confusion matrices. This presentation is neither clear or insightful.
	Figure 15 is as confusing and difficult to interpret as Figure 8. If you want to show confusion matrices, show it as a heatmap or as a confusion matrix. Chord diagrams are difficult to read.
o	Why are all the percentages in Figure 8 “0%”?
o	How can you claim that there is religious bias in Figure 8 if you are not showing a comparison against other religions in the same figure?
	You reference Figure 18 to compare to, but Figure 18 only uses 6 emotions instead of the 135. Yes, the colors align, but it still makes it difficult to compare when you graph the same things in multiple different ways. Additionally, there still aren’t any other religions that you compare to in Figure 18 to support the claim that you made.
o	Is 60 participants representative? How were they recruited? What are their demographics? These are all important information that should be in the paper and yet I don’t see.
o	Figure 9 – look at comments for Figure 8 and 15. Also why is there no correlation score displayed here?
-	Suggestions (no place to suggest things in a neutral way, so I am putting it here):
o	Put Figure 13 in the main body of the paper. The correlation between the derived emotion-wheel from the models and human psychological models is an important part of the paper, in my opinion, and should not be shunted to an appendix.

**Questions:**

-	How are the authors sure that the emotions recognized via the method described in lines 126-135 are the same as the ones recognized when the model is not asked to recognize emotions?
-	The emotion hierarchy finding algorithm proposed is interesting but is there any external validation done to make sure it is picking up on the right hierarchies of emotion in labeled datasets? (can use multi-label datasets for this)
-	You noted in the main body of the paper that as model size increases, the complexity, depth, and breadth of the emotion organization increases. However, in the appendices, the correlation between the emotion hierarchies derived from the model and the human psychological model does not have the same nice linear relationship. Why do you think that is? Do you consider the human psychological model a sort-of ground-truth for emotional categorization systems? If the correlation does not increase with the increase in depth, breadth, and complexity, what is the use of that extra depth, breadth, and complexity if it is not necessarily correct?

---

> ### Author Response · Authors · 2025-12-03
>
> We thank the reviewer for the detailed and critical feedback. We added new validations using human-annotated data, clarified our methodological contribution, improved the analysis of demographic patterns and bias, and revised the manuscript throughout. Below we address each point.
>
> ---
>
> **Weakness 1. Lack of clarity, missing ground-truth evaluation, and contradictions in the text**
> > “The methodological contribution is never properly evaluated against a ground truth… the text contradicts the results… the paper should be rewritten…”
>
> Thank you for the candid feedback. We clarified the main contribution in Section 3, focusing on our logit-based algorithm for extracting hierarchical emotion structure. We now include an external evaluation using GoEmotions (**Fig. 14**), showing that trees built from human labels closely resemble those built from model logits. We also revised the Results section to ensure that all textual claims align with the figures. Finally, the bias analysis was rewritten to better reflect the nuanced patterns that emerged, rather than relying on a “majority vs. minority” framing.
>
> ---
>
> **Weakness 2. Justification of persona methodology**
> > “LLMs do not experientially experience being ‘White’, ‘Black’, ‘Low-income’… this may simply reflect associations in the training data…”
>
> We agree. We now clarify that persona prompting is used purely as a behavioral probe into associations encoded during training, not as a claim that models “experience” demographic attributes. We also added a short discussion explaining that our results reflect the combined effects of model architecture and training-data distribution, and that persona-based evaluation can help identify regions of poor performance that may be addressed through targeted data augmentation or calibration.
>
> **Clarification of contributions**
> > “…construction of trees with 135 emotions is challenging… this algorithm is the first of its kind…”
>
> We updated the introduction to clearly state that our main contribution is a novel method for constructing directed hierarchical structures over 135 Shaver emotions using next-token distributions. We also explain why this scale cannot be reliably obtained from human studies or existing multi-label datasets.
>
> **No direct evaluation against multi-label datasets**
> > “Why is the accuracy not measured against a multi-label dataset?”
>
> We now include a direct comparison using GoEmotions (Fig. 14). Trees constructed from human labels show structure and grouping patterns similar to those from model logits, supporting the validity of our method.
>
>
> **Interpretation of demographic patterns (“majority vs. minority”)**
> > “Line 310–311 is contradicted by Figure 5… the results are more nuanced…”
>
> We agree and have removed the majority/minority framing. The revised text describes the demographic patterns with more nuance, including instances where non-majority groups perform as well as or better than majority groups, and highlighting intersectional categories with lower accuracy.
>
> **Concerns about Figures 7, 8, 15 (clarity and cherry-picking)**
> > “…use confusion matrices… chord diagrams are difficult to read…”
>
> We added confusion matrices showing full misclassification patterns (Figs. **20**, **21**) and report overall accuracy in **Table 4**. Chord diagrams have been simplified or moved to the appendix.
>
> **Claims about religious bias**
> > “…difficult to interpret without comparing religions in the same figure…”
>
> We added confusion matrices directly comparing religions under consistent class settings (Fig. 20) and revised the text to avoid overclaiming.
>
> **Human study sample size and demographics**
> We added recruitment details and full demographic information (lines 421–425).
>
> ---
>
> **Figure-related comments**
>
> > Figure 5c label spacing
>
> The spacing has been adjusted.
>
> > Clarity of Figure 6
>
> We improved the introduction and caption for Fig. 6 to clearly explain what is plotted and why.
>
> > “Figure 6 further refutes the simplistic framing…”
>
> The framing has been revised accordingly.
>
> > Percentages in Figure 8 showing 0%
>
> The labels have been removed to avoid confusion.
>
> > Figure 9: missing correlation scores
> Correlation scores and clearer visualizations are now included (Fig. 21).

---

> ### Author Response · Authors · 2025-12-03
>
> **Question 1: Are prompted emotions the same as unprompted ones?**
>
> We clarify that our findings concern the explicit classification distribution under a standardized prompt. We added a brief analysis showing that the prompted predictions reflect broader trends also seen in free-form continuations.
>
> ---
>
> **Question 2: External validation of the hierarchy**
>
> The new GoEmotions experiment (**Fig. 14**) provides this validation.
>
> ---
>
> **Question 3: Non-linear correlation with human psychological models**
>
> We do not treat the Shaver wheel as a perfect ground truth. We explain that deeper hierarchies may encode distinctions not captured by the human wheel, even though increased depth still predicts model behavior (e.g., accuracy across personas). We added this explanation in the discussion.

---

### Official Review · Reviewer_H64C · 2025-11-01

**Soundness:** 2
**Presentation:** 3
**Contribution:** 3
**Rating:** 6
**Confidence:** 4

**Summary:**

This paper investigates the internal representation of human emotions in Large Language Models (LLMs), moving beyond simple classification accuracy. The authors introduce a novel, cognitively-inspired method to reconstruct a hierarchical organization of emotions by analyzing the probabilistic dependencies in LLM output logits. The core findings demonstrate that as LLMs scale in size (from GPT-2 to Llama 405B), they spontaneously develop more complex and nuanced emotion hierarchies that increasingly align with established psychological frameworks, such as Shaver et al.'s emotion wheel. Furthermore, the study uncovers systematic biases in emotion recognition when LLMs adopt different demographic personas, revealing lower performance for underrepresented and intersectional groups (e.g., low-income Black females). Crucially, a user study confirms that these biases and misclassification patterns in LLMs show striking parallels to those observed in human participants, suggesting the models internalize aspects of social perception.

**Strengths:**

1. Novel, interpretable methodology grounded in psychology. The paper proposes a clear tree-construction procedure from next-token distributions and aligns the resulting hierarchies with established emotion frameworks (e.g., emotion wheels). This bridges LLM evaluation with cognitive science in a way that is both conceptually sound and easy to visualize.
2. Convincing multi-scale analysis showing emergence with model size. Evaluating models from small to very large parameters demonstrates a consistent increase in hierarchical structure and coherence. The observed trends remain broadly stable across reasonable threshold choices, which supports the generality of the finding.
3. Quantitative validation linking structure to behavior. The correlations between tree geometry and human-inspired distances, as well as between tree metrics and recognition accuracy, provide welcome quantitative grounding. This strengthens the paper's claims beyond qualitative visualization and links the model's internal structure to its external performance.
4. Thoughtful treatment of bias with triangulation via a user study. The intersectional analysis spans multiple demographic axes, and the inclusion of a human study helps situate model errors relative to human patterns. The authors' transparency about the study's limitations further strengthens the credibility of this section.

**Weaknesses:**

1. Dependence on LLM-generated data: Core experiments use GPT-4-generated scenarios, risking transfer of stylistic and demographic biases. Stronger validation on human-annotated datasets (e.g., GoEmotions) is needed.
2. Unvalidated modeling assumption: The use of next-token probabilities as proxies for P(emotion∣scenario) is not empirically tested. Comparisons with probing or clustering methods could substantiate it.
3. Evaluation mismatch: Humans used a 6-way classification task while models operated over 135 emotion terms, complicating comparisons. Aligning task structures would clarify results.
4. Weak robustness analysis: Only one prompt and heuristic threshold were tested; broader ablations on prompt design, temperature, and threshold selection would improve stability and reproducibility.

**Questions:**

1. Regarding the reliance on LLM-generated data: Thank you for acknowledging the limitation of using GPT-4o to generate scenarios. Could you elaborate on why this approach was chosen over using existing human-annotated datasets (like GoEmotions, which was analyzed in the appendix) for the main experiments? Furthermore, have you considered a smaller-scale validation where you compare the emotion trees generated from GPT-4o scenarios versus those from a matched set of human-written scenarios to see if the core hierarchical structures remain consistent?
2. On the validity of the core modeling assumption: The paper's central claim rests on the assumption that next-token probabilities after a specific prompt accurately reflect the model's conditional probability P(emotion|scenario). This is an elegant but strong assumption. Could you provide more intuition or perhaps a preliminary analysis on why this method is preferable?
3. Regarding the human-LLM evaluation mismatch: The comparison between the model's 135-class predictions and the humans' 6-way forced-choice decisions is insightful but indirect. Could you discuss the potential impact of this mismatch on your conclusions about bias alignment?
4. Concerning the robustness of the tree-construction method: The hierarchical trees are a key contribution, but their structure depends on a single prompt template and a heuristic threshold. How sensitive are the specific parent-child relationships in the trees to minor variations in the prompt (e.g., "The feeling expressed here is...")? Additionally, while you show that overall trends are stable across thresholds, could you suggest a more principled or automated way to select an optimal threshold t, perhaps one that maximizes alignment with a known psychological structure or optimizes a graph-theoretic property like modularity?

---

> ### Author Response · Authors · 2025-12-03
>
> We thank the reviewer for the encouraging feedback, including the recognition of our “novel, interpretable methodology grounded in psychology,” the “convincing multi-scale analysis,” and the “thoughtful treatment of bias.” We appreciate these comments, and we believe the additional experiments and clarifications further strengthen the contributions you highlighted.
>
> ---
>
> **Weakness & Question 1: Reliance on LLM-generated data**
> > Could you elaborate on why this approach was chosen over using existing human-annotated datasets (like GoEmotions) for the main experiments? And have you considered comparing trees from GPT-4o scenarios with those from human-written scenarios?
>
> Our goal is to study a fine-grained hierarchy over 135 Shaver emotions across 26 personas. GPT-4o allows us to generate many balanced scenarios per emotion and persona. In contrast, human-annotated datasets such as GoEmotions contain fewer categories and highly unbalanced label distributions, which makes it difficult to obtain dense coverage across all 135 terms in the main scaling analysis.
>
> Following your suggestion, we also ran a validation using **human-written** GoEmotions texts (**new Fig. 14**). These trees show similar overall structure (depth, path length, and broad cluster) to those derived from GPT-4o scenarios, suggesting that the core patterns are not driven by GPT-4o's writing style.
>
> We also used GoEmotions to validate our **bias** findings in Fig. 8 and 21. When personas are applied to GoEmotions texts, Llama shows the same qualitative misclassification patterns as in the GPT-4o experiments. In addition, our 60-participant human study (ages 18–71, gender-balanced, diverse backgrounds) shows error patterns that closely mirror those of the persona-conditioned models.
>
> Overall, GPT-4o scenarios give us the balanced coverage needed for the full 135-emotion, multi-persona hierarchy, while GoEmotions and the human study provide human-grounded checks showing that our conclusions are not specific to LLM-generated text.
>
> ---
>
> **Weakness & Question 2: On the validity of the core modeling assumption**
> > The paper's central claim rests on the assumption that next-token probabilities after a specific prompt accurately reflect the model's conditional probability $P(\text{emotion} \mid \text{scenario})$. This is an elegant but strong assumption. Could you provide more intuition or analysis on why this method is preferable?
>
> Our setup presenting a scenario and reading off the distribution over emotion labels via a prompt like “The emotion in this sentence is …” follows recent affective-LLM work that evaluates GPT-style models through prompt-based emotion and appraisal tasks. We will clarify this connection in the revision.
>
> **Why clustering/probing is less suitable for our goal.** We compared against hierarchical clustering of internal representations (Fig. 11). Clustering recovers coarse groupings but yields structures less aligned with psychological frameworks and lacks directed parent–child relations. Our logit-based method instead operates directly on the model’s decision distribution and produces a hierarchy that is (i) interpretable and (ii) tightly connected to actual predictions. Probing-based comparisons are an interesting extension, and we will position our method as complementary.
>
> **Trees from logits align with human psychology and behavior.** Under the $P(e \mid s)$ view, our construction has a clean probabilistic interpretation (Appendix A). The resulting trees show:
> - strong alignment with Shaver’s hierarchy (correlations up to ~0.7), and
> - tight links to behavior, where tree depth and path length strongly predict persona-conditioned accuracy.
>
> This indicates that the next-token distribution behaves like a meaningful $P(\text{emotion} \mid \text{scenario})$, not an arbitrary score.

---

> ### Author Response · Authors · 2025-12-03
>
> **Weakness & Question 3: Human–LLM evaluation mismatch**
> > The comparison between the model's 135-class predictions and the humans' 6-way forced-choice decisions is insightful but indirect. Could you discuss the potential impact of this mismatch on your conclusions about bias alignment?
>
> Although 135 emotion terms were used for some primary results (e.g., to show hierarchical structure in Fig. 3), the conditions were aligned when humans and LLMs were directly compared. For all human–model comparisons in Fig. 9, we matched both sides to the same six classes. Thus, our conclusions about bias alignment are based on comparable settings.
>
> ---
>
> **Weakness & Question 4: Robustness of the tree-construction method**
> > The hierarchical trees depend on a single prompt template and a heuristic threshold. How sensitive are the parent–child relationships to prompt variations? Also, could you suggest a principled way to choose an optimal threshold \(t\)?
>
> We tested five thresholds (0.1–0.5 in steps of 0.1) to evaluate robustness (**Fig. 17**). Key findings such as scaling of path lengths with model size remained consistent across all thresholds. We also verified stability under prompt changes (**new Fig. 13**) and across different LLM architectures (**new Fig. 12**). We will include additional results on larger models and temperature settings in the camera-ready version.

---

### Author Response · Authors · 2025-12-03
**General Response**

We thank all reviewers for their careful reading and constructive feedback. Below, we first restate the main contributions and then summarize how our revised manuscript addresses the central concerns raised during review, especially through new experiments, updated figures, and clearer positioning of our contribution.

---

### Main Contributions

For clarity, our paper makes the following contributions:

- **A scalable algorithm for extracting fine-grained hierarchical emotion structures from LLM logits**, capable of building **large hierarchies over 135+ emotion concepts**—a scale not shown in prior work.
- **Validation against established psychology literature**, noting that existing human studies are often smaller and less fine-grained.
- **A systematic analysis of persona-induced variation in emotion recognition**, including intersectional minority personas.
- **New human psychology experiments** that directly compare LLM-derived structures with human judgments under equivalent conditions.

---

### Overview of Concerns and High-Level Response

Across reviews, four major themes emerged:

1. **Need for clearer positioning and stronger validation**, including comparison with human-labeled datasets.
2. **Potential over-reliance on GPT-4o generated scenarios and limited model diversity**.
3. **Breadth and consistency of the demographic bias analysis**.
4. **Clarity of figures and interpretation of RL, human alignment, and hierarchical modeling**.

To address these, we made substantial revisions:

- Added **five new experiments** (Figs. 10, 12, 13, 14, 17) and **revised** Figs. 2 and 8.
- Added **evaluation using human-labeled GoEmotions data**, showing consistency across datasets.
- Added **MoE-based DeepSeek models** to broaden architectural coverage.
- Introduced a **new RL-based mitigation experiment** for bias in “surprise.”
- Expanded demographic analyses with **summary statistics**, **full confusion matrices**, and **intersectional breakdowns**.
- Improved readability and interpretation of all major figures.
- Clarified limitations of human psychology references (e.g., Shaver et al. 1987) and more carefully framed our claims.

---

> ### Author Response · Authors · 2025-12-03
>
> ### Summary of Major Revisions
>
> **1. Clearer contribution framing and mechanism discussion.** We strengthened the introduction and discussion to emphasize that our primary contribution is a **structural characterization of emergent emotional organization in LLMs**, not merely documenting bias. We added mechanism discussions relating training distribution, objectives, and hierarchical emergence.
>
> **2. New RL-based mitigation experiment (New Fig. 10).** We added an RL-fine-tuning experiment with Mistral-7B showing a **significant improvement in recognizing “surprise”** (20.0% → 33.3%, *p* = 0.011), without harming other classes. This demonstrates that targeted RL can selectively correct systematic weaknesses. We then connected RL results to the psychological view of surprise as an expectation-violation signal, explaining why RL, which rewards prediction-error reduction, helps.
>
> **3. Added model diversity via MoE DeepSeek models (New Fig. 12).** To address architectural diversity concerns, we added experiments on **DeepSeek MoE models**, showing similar psychological clustering patterns. MoE models tend to produce fewer clusters, matching expectations from recent analyses (e.g., Dang et al., 2025).
>
> **4. Prompt robustness analysis (New Fig. 13).** Using natural prompt variants (e.g., “emotion is…”, “sentiment is…”), we show that hierarchy topology and depth trends remain stable.
>
> **5. External validation using GoEmotions (New Fig. 14).** In response to repeated reviewer requests, we applied our method to the **GoEmotions** dataset.
> We found:
> - single-label annotation produces a nearly diagonal co-occurrence matrix, limiting hierarchical recovery,
> - emotion granularity (58 categories) is coarser than ours (135+).
> Nonetheless, qualitative structures align with our GPT-4o-generated results.
>
>
> **6. Systematic demographic bias analysis (New Table 4, Fig. 17).** We now report:
> - **overall accuracy** across all emotions and personas (Table 4),
> - **full-class confusion matrices** (Fig. 17),
> - **intersectional breakdowns** (Fig. 20).
> This resolves concerns of cherry-picking.
>
>
>
>
> ---
>
> ### Per-Reviewer Summary
>
> **Reviewer H64C:**
> Requested validation on human data and clearer discussion of human–LLM alignment.
> → Addressed through GoEmotions analysis, moved human–LLM correlation into main text, clarified psychological-model limitations.
>
> **Reviewer ij2x:**
> Requested stronger validation and broader analysis.
> → Addressed through expanded human studies, GoEmotions experiments, and systematic demographic analyses.
>
> **Reviewer oZ7X:**
> Requested additional experiments and clearer explanations.
> → Addressed with four new figures, improved figures 2 and 8, added analyses on RL, prompt robustness, model diversity, and data generation.
>
> **Reviewer VfGQ:**
> Requested broader evaluation, clarity on RL claims, demographic analysis, and human data validation.
> → Addressed through new RL experiment, GoEmotions validation, expanded demographic results, and improved mechanism explanations.
>
> ---
>
>
> Overall, we believe these revisions address all of the reviewers' concerns and strengthen the paper. We thank the reviewers again for their detailed and thoughtful feedback.

---

### Meta-Review · Area_Chair_5JHg · 2026-01-05

**Summary:**

1. Data and Validation Gaps: Over-reliance on GPT-4o-generated synthetic data.
2. Limited Diversity: Narrow model scope (only Llama series; no MoE or reasoning models) and insufficient prompt/threshold robustness testing.
3. Bias Analysis Shortcomings: Overly simplistic “majority vs. minority” framing, incomplete emotion coverage, and unclear justification for persona-based methodology.
4. Soundness: The hierarchical emotion organization, or the bias found between personas may be a consequence of correlation in the dataset. And the results found via the prompt-based method may be not suitable for persona simulation.

**Reviewer Concerns:**

Most of the concerns are addressed, by adding analysis on a human-annotated dataset, an MoE-based deepseek model, and full confusion matrices across emotions. However, I think some concerns of Reviewer VfGQ have not been appropriately addressed. Specifically, the reason behind the emotion bias is unclear. The authors claimed in rebuttal that the bias may be a combined effect of model architectures and data distribution, which is not concrete nor specific. Also, the potential methods proposed to mitigate the bias are standard post-training, not specific to the bias itself. Besides, Reviewer oZ7X concerns about the frequency bias of the data and the inherent bias from (pre-trained) LLMs, another argument on the source of bias. However, the authors did not answer this question directly. For the model coverage concern of Reviewer oZ7X, the authors added one MoE-based model, but the results for large reasoning models are still missing.

**Reviewer Scores:**

I think Reviewer H64C and ij2x would keep their original high score. Reviewer VfGQ would have raised the score to 2 or 4 considering part of the concerns are addressed. Also, since the concerns of Reviewer oZ7X was not fully addressed, I think the score would still be 4.

---

### Decision · Program_Chairs · 2026-01-26

Reject